# Genome rearrangements and pervasive meiotic drive cause hybrid infertility in fission yeast

Sarah E Zanders[1], Michael T Eickbush[1], Jonathan S Yu[1†], Ji-Won Kang[1,3], Kyle R Fowler[1], Gerald R Smith[1], Harmit Singh Malik[1,2]*

[1]Division of Basic Sciences, Fred Hutchinson Cancer Research Center, Seattle, United States; [2]Howard Hughes Medical Institute, Fred Hutchinson Cancer Research Center, Seattle, United States; [3]University of Washington, Seattle, United States

**Abstract** Hybrid sterility is one of the earliest postzygotic isolating mechanisms to evolve between two recently diverged species. Here we identify causes underlying hybrid infertility of two recently diverged fission yeast species *Schizosaccharomyces pombe* and *S. kambucha*, which mate to form viable hybrid diploids that efficiently complete meiosis, but generate few viable gametes. We find that chromosomal rearrangements and related recombination defects are major but not sole causes of hybrid infertility. At least three distinct meiotic drive alleles, one on each *S. kambucha* chromosome, independently contribute to hybrid infertility by causing nonrandom spore death. Two of these driving loci are linked by a chromosomal translocation and thus constitute a novel type of paired meiotic drive complex. Our study reveals how quickly multiple barriers to fertility can arise. In addition, it provides further support for models in which genetic conflicts, such as those caused by meiotic drive alleles, can drive speciation.

*For correspondence: hsmalik@fhcrc.org

Present address: †Yale University, New Haven, United States

Competing interests: The authors declare that no competing interests exist.

Reviewing editor: Detlef Weigel

## Introduction

Identifying the molecular and evolutionary bases of hybrid sterility is necessary for understanding the mechanisms of speciation. Hybrid sterility is one of the earliest reproductive isolation mechanisms to evolve between two recently diverged species (*Coyne and Orr, 2004*), yet we are only beginning to understand the types of genetic changes that lead to hybrid infertility (*Coyne and Orr, 2004*; *Johnson, 2010*; *Presgraves, 2010*). Since the evolutionary forces driving genetic changes that cause infertility between species are likely also acting within species, the study of hybrid sterility also promises significant insight into mechanisms underlying infertility within species.

The (Bateson) Dobzhansky-Muller (BDM) model provided a solution to the paradox of how genetic changes that lead to speciation could be tolerated by natural selection despite decreasing the fitness potential of an organism. This model proposes that hybrid sterility results from incompatibilities between genes that evolved in different populations and were therefore never tested together by natural selection (*Coyne and Orr, 2004*). Indeed, incompatible BDM pairs have been identified in diverse organisms that either cause hybrid sterility or reinforce species isolation (*Brideau et al., 2006*; *Lee et al., 2008*; *Bayes and Malik, 2009*). Although relatively few loci underlying hybrid incompatibilities have been identified, one theme that has emerged is that the loci are often rapidly evolving and implicated as players in 'molecular evolutionary arms races'. These arms races can occur between host genomes and external forces such as parasites (*Bomblies et al., 2007*). Alternatively, the genetic conflicts can be between different elements *within* a genome, such as between selfish parasitic genes and other host genes (*Johnson, 2010*; *Presgraves, 2010*).

**eLife digest** It is widely thought that all of the billions of species on Earth are descended from a common ancestor. New species are created via a process called speciation, and nature employs various 'barriers' to keep closely related species distinct from one another. One of these barriers is called hybrid sterility. Horses and donkeys, for example, can mate to produce hybrids called mules, but mules cannot produce offspring of their own because they are infertile.

Hybrid sterility can occur for a number of reasons. Mules are infertile because they inherit 32 chromosomes from their horse parent, but only 31 chromosomes from their donkey parent—and so have an odd chromosome that they cannot pair-off when they make sperm or egg cells. However, even if a hybrid inherits the same number of chromosomes from each parent, if the chromosomes from the two parents have different structures, the hybrid may still be infertile.

Zanders et al. have now looked at two species of fission yeast—*S. pombe* and *S. kambucha*—that share 99.5% of their DNA sequence. Although hybrids of these two species inherit three chromosomes from each parent, the majority of spores (the yeast equivalent of sperm) that these hybrids produce fail to develop into new yeast cells. Zanders et al. identified two causes of this infertility: one of these was chromosomal rearrangement; the other was due to three different sites in the DNA of *S. kambucha* that interfere with the development of the spores that inherit *S. pombe* chromosomes.

Since these two yeast species are so closely related, the findings of Zanders et al. reveal how quickly multiple barriers to fertility can arise. In addition, these findings provide further support for models in which conflicts between different genes in genomes can drive the process of speciation.

Despite their explanatory power, DM incompatibilities are not exclusive causes of hybrid infertility. For instance, changes in ploidy are a rapid means of speciation in plants (*Otto and Whitton, 2000*). Defects in meiotic recombination contribute to hybrid infertility in both mouse and budding yeast hybrids (*Hunter et al., 1996*; *Bhattacharyya et al., 2013*; *Mihola et al., 2009*). In addition, genomic rearrangements can also cause or contribute to speciation (*White, 1978*; *Faria and Navarro, 2010*; *Hoffmann and Rieseberg, 2008*; *Noor et al., 2001*). In the classic chromosomal speciation model, chromosomal rearrangements between populations lead to infertility when heterozygous. Like DM gene incompatibilities, chromosomal rearrangements can contribute to hybrid infertility and serve as a genetic barrier between populations (*White, 1978*). For example, the transposition of an essential fertility gene causes male infertility in some *Drosophila* hybrids and chromosomal rearrangements contribute to hybrid infertility in some budding yeast hybrids (*Masly et al., 2006*; *Delneri et al., 2003*).

How do chromosomal rearrangements become established in organisms in which they cause infertility when heterozygous? One possibility is that a rearrangement could become fixed in a small population via genetic drift and inbreeding (*Rieseberg, 2001*). White proposed an alternative solution in which novel chromosomal rearrangements could increase in frequency if they were linked to meiotic drive alleles (*White, 1978*). These selfish genetic elements 'cheat' to be transmitted to more than 50% of the functional gametes of a heterozygote (*Burt and Trivers, 2006*). Due to their transmission advantage, meiotic drive alleles and loci linked to them can spread through a population even if they cause fertility decreases (*Crow, 1991*). In this way, even a chromosomal rearrangement that causes decreased fertility when heterozygous could become fixed in a population if it is linked to a strong meiotic drive allele. Because loci linked to drive alleles also benefit from the transmission advantage, linked variants that enhance drive will also be selected (*Crow, 1991*). Chromosome inversions that prevent recombination thereby reinforcing the linkage between drive alleles and their enhancers can also spread through a population due to enhanced drive. In this way, meiotic drive alleles can even *promote the evolution* of chromosomal rearrangements, in spite of their fitness costs. Consistent with this, meiotic drive loci are commonly found within inversions and have been proposed to underlie other types of dramatic karyotype evolution (*Dyer et al., 2007*; *Larracuente and Presgraves, 2012*; *Pardo-Manuel de Villena and Sapienza, 2001*; *Hammer et al., 1989*). However, there is currently little experimental or theoretical support for White's model; genetic conflicts and chromosomal rearrangements are still considered distinct causes of hybrid sterility and speciation.

Here, we show that a combination of selfish meiotic drive loci and chromosomal rearrangements leads to near complete hybrid sterility between two recently diverged fission yeast species. This study

provides support for models in which selfish genetic elements and chromosomal rearrangements are drivers of hybrid dysfunction (*Johnson, 2010*; *Presgraves, 2010*; *White, 1978*). Our observations in fission yeast are consistent with White's chromosomal speciation model.

## Results

### *Sk/Sp* hybrids have low spore viability

*S. pombe* (*Sp*) and *S. kambucha* (*Sk*) generally exist as single-celled haploids, but cells of opposite mating types readily mate to form single-celled diploids. Fission yeasts are homothallic; they can switch mating types during mitotic growth and self-mate effectively. Although there is evidence of genetic outcrossing in yeasts closely related to *Sp*, the relative frequencies of outcrossing vs selfing are unknown (*Brown et al., 2011*). *Sk* and *Sp* are 99.5% identical at the DNA sequence level (*Rhind et al., 2011*). Despite their genetic similarity, *Sk/Sp* hybrid diploids are mostly infertile (*Singh and Klar, 2002*). To investigate the causes of the hybrid infertility, we generated a suite of genetic markers ('Materials and methods') that transformed *Sk* from a non-model yeast into a distinct genetically trac-table model system. In addition to their use in this study of fertility, these tools will also facilitate molecular dissection of the functional consequences of evolution between fission yeast species.

*Sk/Sp* hybrid diploids displayed no obvious mitotic defects, indicating there are no dominant lethal incompatibilities between the two species that act during mitosis (*Figure 1A,B*). To assay hybrid fertility, we measured the viable spore yield of *Sk/Sp* hybrids, *Sk/Sk*, and *Sp/Sp* diploids (*Smith, 2009*). This assay measures the number of viable spores (gametes) produced per viable diploid placed on the starvation medium that induces cells to undergo meiosis. Spores are considered viable if they are able to grow into a visible colony. Because cells can undergo a few mitotic divisions prior to meiosis and not all spores can be recovered from the starvation medium, the assay is a relative, rather than an absolute, measure. *Sk/Sk* diploids had a slightly lower viable spore yield than *Sp/Sp* diploids, 3.6 vs 8.4 (unpaired *t* test p=0.053; *Figure 1C*). However, when assayed via micromanipulation of individual spores, spores derived from *Sp/Sp* and *Sk/Sk* diploids were equally viable. We therefore conclude that the difference between *Sp/Sp* and *Sk/Sk* diploids in the viable spore yield assay is due to an extra mitosis of *Sp/Sp* diploids before meiotic induction, not higher viability of gametes. In sharp contrast, *Sk/Sp* hybrids had a viable spore yield at least 24-fold less than that of either of the pure species diploids (*t* test p<0.01; *Figure 1C*).

We considered whether the low fertility of *Sk/Sp* hybrids reflected a defect in initiating or complet-ing meiosis. To test this possibility, we monitored meiotic progression through a time course. Using DAPI staining of cells, we found that *Sk/Sp* hybrids initiated and completed both meiotic nuclear divi-sions with similar timing and efficiency compared to *Sp/Sp* and *Sk/Sk* diploids (*Figure 1D*). These results suggest that spore inviability in products of hybrid meiosis is not due to an inability to enter meiosis or due to checkpoint activation preventing the completion of the meiotic divisions. However, we did find that the spores produced by *Sk/Sp* hybrids were more irregular and less refractile than those produced by pure species diploids (*Figure 1E,F*).

Furthermore, the viable spores produced by *Sk/Sp* hybrids grew into colonies that were grossly different from those of pure species controls (*Figure 1G*). There were a few large colonies from the hybrid spores, but most were small and many had irregularly shaped edges, rather than large colonies with smooth circular perimeters like those produced by pure species spores. When the small hybrid spore colonies were streaked for single colonies on fresh Petri plates, some cells within the colony lost the slow growth phenotype and generated colonies similar in size to those of *Sp* or *Sk* haploids.

These phenotypes are consistent with the small colonies being chromosome 3 aneuploids (disomes); aneuploidy for chromosome 1 or 2 is lethal in *Sp* (*Niwa et al., 2006*). To formally test this idea, we measured the frequency of heterozygous aneuploids amongst viable spores using co-dominant heter-ozygous markers at allelic loci (one on the *Sk* chromosome and one on the *Sp* chromosome). Spores that inherited both markers on chromosome 2 were heterozygous diploids. Spores that inherited both markers on chromosome 3, but only one marker from chromosome 2, were heterozygous aneuploids. Homozygous diploids and aneuploids were not detectable with the markers used here and were counted as haploids. From *Sk/Sp* hybrids, 33% of the viable spores were heterozygous aneuploids and 44% were heterozygous diploid, significantly more than the 2.2 and 3.5%, respectively, observed in *Sk* control diploids (*Figure 1H*; p<0.01 for both; G-test).

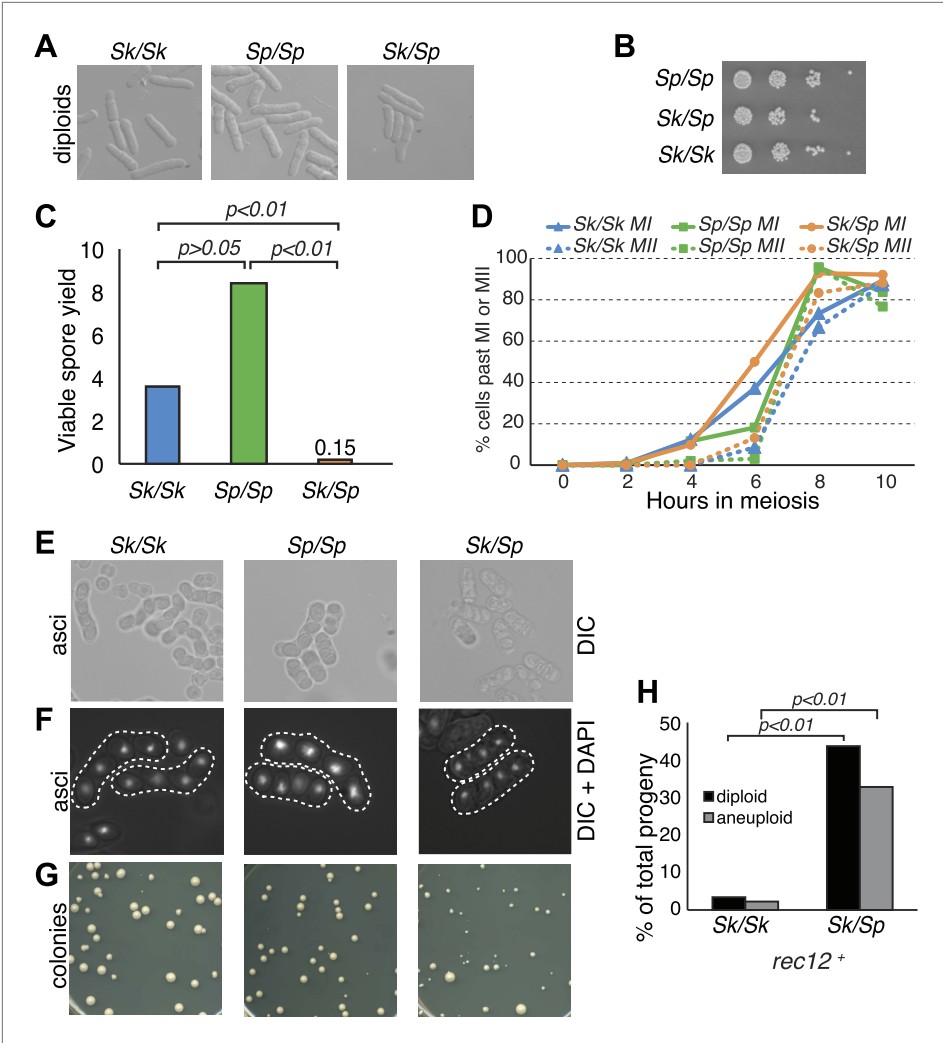

**Figure 1**. *Sk/Sp* hybrids are healthy but exhibit low fertility. (**A**) *Sk/Sp* hybrid diploids are morphologically similar to pure species diploids. (**B**) *Sk/Sp* hybrid diploids show no gross growth defects relative to pure species controls. (**C**) Viable spore yield tests show that *Sk/Sp* fertility is low relative to pure species controls (averages of n ≥ 5 experiments, p-values obtained using *t* test). This assay does not directly measure viable spores per meiosis, so values can exceed 4. (**D**) *Sk/Sp* hybrids complete both meiotic divisions with timing similar to that of pure species controls (representative experiment of 3, n ≥200 cells for each data point). (**E** and **F**) The asci produced by *Sk/Sp* hybrids contain spores that are more irregular and transparent than pure species asci. (**G**) The viable spores produced by *Sk/Sp* hybrids often grow into small irregularly sized and shaped colonies. (**H**) The majority of the viable spores produced by *Sk/Sp* hybrids are aneuploid or diploid (p-values obtained using G-test, n >200 for each). These data are also shown in *Figure 5A*.

## Meiotic double-strand DNA break formation and repair appears normal in *Sk/Sp* hybrids

One possible explanation for the low fertility and the high frequency of aneuploids and diploids we observed in *Sk/Sp* hybrids, is that these hybrids may have defects in meiotic recombination. In many eukaryotes, meiotic recombination is essential for the production of viable gametes. This is because recombination can form crossovers, which help ensure proper disjunction of homologous chromosome pairs during the first meiotic division (*Kerr et al., 2012*). When a pair of homologous chromosomes fails to form a crossover in meiosis, chromosomes segregate less faithfully and can produce aneuploid gametes. In both budding yeast and mouse hybrids, defects in meiotic recombination are major causes of hybrid infertility (*Hunter et al., 1996*; *Bhattacharyya et al., 2013*; *Mihola et al., 2009*).

These observations, combined with rapid evolution of genes that function in meiotic recombination, have led to the hypothesis that recombination defects serve as a general barrier to genetic exchange between species (*Rhind et al., 2011*; *Anderson et al., 2009*; *Sawyer and Malik, 2006*; *Myers et al., 2010*). We therefore tested whether meiotic recombination is aberrant in *Sp/Sk* hybrids.

Meiotic recombination is initiated by double-strand DNA breaks (DSBs) induced by a complex of proteins including the conserved Spo11 protein (Rec12 in fission yeast). In many organisms, including *Sp*, meiotic DSBs are not randomly distributed; rather they are concentrated in regions called hotspots (*Cromie et al., 2007*). DSB hotspots are thought to evolve rapidly because they can be lost due to biased gene conversion during break repair (*Boulton et al., 1997*). Perhaps related to this, rapid divergence of a DSB hotspot-determining protein contributes to infertility of certain mouse hybrids and some human males (*Mihola et al., 2009*; *Myers et al., 2010*; *Segurel et al., 2011*; *Baudat et al., 2010*; *Irie et al., 2009*). To test if DSB hotspot divergence contributes to *Sk/Sp* hybrid infertility, we mapped the locations of *Sk* DSB hotspots. Rec12 remains covalently linked to the DNA it cuts, so DSB sites can be mapped by identifying the DNA linked to Rec12 (*Keeney et al., 1997*). To do this in *Sk*, we first generated a strain containing a FLAG-tagged *rec12* gene and the *rad50S* mutation in which DSBs form normally but are not repaired (*Alani et al., 1990*; *Hyppa et al., 2008*). We then used chromatin immunoprecipitation on meiotic extracts to pull down Rec12-FLAG and the covalently attached DNA. We identified the bound DNA by amplification and hybridization to a microarray. Qualitatively, we found that the overall genome-wide pattern of DSB hotspots in *Sk* was similar to the published *Sp* hotspot map (*Figure 2—figure supplement 1*; *Cromie et al., 2007*). We then compared the DSB hotspots between *Sk* and *Sp* at 286 previously identified *Sp* hotspots and found strong correlation between hotspot intensities in the two species (*Figure 2A*; *Fowler et al., 2013*). This result shows that hotspots have largely not diverged between *Sp* and *Sk*, so such divergence is unlikely to contribute to hybrid infertility.

A second hypothesis is that hybrids have gross defects in forming or repairing meiotic DSBs, perhaps due to incompatibilities in the meiotic recombination machinery. To observe DSB formation, we used pulsed-field gel electrophoresis (PFGE) of whole chromosomes in *rad50S* hybrid and control diploids. We found that like those of pure species, the *Sk/Sp* chromosomes were effectively broken in meiosis (*Figure 2B*). For a more precise view of DSB formation in hybrids, we assayed break formation in *rad50S* diploids at known meiotic DSB hotspots contained on the *Not*I restriction fragments designated 'J' (0.5 Mb) and 'D' (1.1 Mb) of the *Sp* genome (*Cromie et al., 2007*). We found that both *Sk* and hybrid diploids exhibited similar amounts of breakage at the same sites, in accord with the microarray results above (*Figure 2C,D*). To assay DSB repair, we visualized meiotic chromosomes via PFGE in *rad50+* (DSB repair proficient) diploids. As in pure species, chromosomes of *Sk/Sp* diploids were largely intact throughout meiotic prophase (*Figure 2E*). This shows that the breaks that form (e.g., *Figure 2B*) are efficiently repaired. Together, these results suggest that DSB formation is grossly normal in *Sk/Sp* hybrids and that the breaks are repaired efficiently.

Meiotic recombination promotes fertility but is not essential for producing viable spores in *Sp* (*Davis and Smith, 2003*). This is largely because random segregation of three chromosome pairs at the first meiotic division would frequently (~25% of the time) result in a viable chromosome complement: haploid, diploid, or chromosome 3 disome. In addition, *Sp*, like some other species, has a recombination-independent chromosome segregation system that is partially effective in promoting proper chromosome segregation (*Davis and Smith, 2003*). As expected, when we eliminated nearly all meiotic recombination by deleting *rec12* (*rec12Δ*), both *Sp/Sp* and *Sk/Sk rec12Δ* mutant diploids had viable spore yields fourfold to fivefold lower than the corresponding *rec12+* diploids (*Figure 3*). Surprisingly, a similar decrease in fertility was not observed in *rec12Δ Sk/Sp* hybrid diploids: their viable spore yield was indistinguishable from that of *rec12+* hybrid diploids (*Figure 3*). This result demonstrates that recombination does not promote *Sk/Sp* hybrid fertility, unlike in pure species. This could be because *Sk/Sp* hybrids fail to form crossovers and/or because the recombination events that occur in hybrids hurt fertility just as much as they promote it.

## Chromosomal rearrangements contribute to *Sk/Sp* hybrid infertility

We next compared recombination frequencies in meioses from either parental pure-species or *Sk/Sp* hybrid diploids. We found that recombination frequencies in *Sk/Sk* were lower than those in *Sp/Sp* in at least two genetic intervals (*Figure 4—figure supplement 1*; *Young et al., 2002*). We also observed more dramatic decreases in observed recombinant frequencies in several genetic intervals amongst

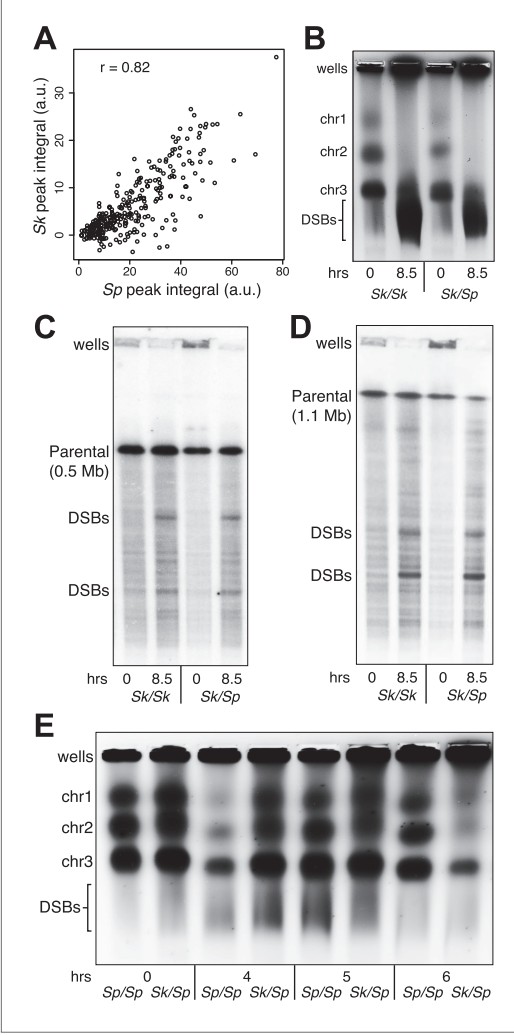

**Figure 2**. DSB hotspot divergence and repair in *Sk/Sp* hybrids. (**A**) We used ChIP–chip of Rec12-FLAG from *rad50S Sk* meiotic cultures to assay DSB hotspots and compared the profile to the published DSB hotspot maps of *Sp* (*Fowler et al., 2013*). We then compared the Rec12-enrichment in *Sk* at 286 defined *Sp* hotspots and found a strong correlation between Rec12 enrichments between the two species at these sites. (**B**) *Sk/Sp* cells are proficient at inducing DSBs. Ethidium bromide stained pulsed-field gel of diploids at 0 and 8.5 hr after inducing meiosis in liquid cultures. These diploids are *rad50S* mutants, so DSBs form normally but are not repaired. As DSBs are formed, the three full-sized chromosome bands disappear and the DNA runs as smaller broken fragments on the gel. (**C** and **D**) We find that DSBs are formed at similar locations and similar frequencies in *Sk/Sp* and *Sk/Sk*. Southern blots of pulsed-field gels to obtain a closer view of DSB formation in *rad50S* diploids probed to visualize two NotI restriction fragments known as NotI J [shown in (**C**)] and NotI D [shown in (**D**)]. Prior to DSB formation, most of the DNA runs as a single large band. After all break formation (8.5 hr) smaller cut fragments become
*Figure 2. Continued on next page*

viable spores produced by *Sk/Sp* hybrids compared to *Sp/Sp* (*Figure 4A*, *Figure 4—figure supplements 2, 3*). The magnitude of the decrease was greatest in the *lys1-lys7* interval where we observed >40-fold decrease in genetic distance in hybrids compared to *Sp/Sp*.

Large decreases in recombination could be explained if some DSBs are infrequently repaired as crossovers in hybrids, or if the crossovers they produce generate inviable chromosomes, for instance due to chromosomal rearrangements. To address this latter possibility, we resequenced the *Sk* genome ('Materials and methods') (*Rhind et al., 2011*), which revealed that a large region of chromosome 1 (between base pairs 2,683,632 and 4,911,515) is inverted in the *Sp* genome, relative to *Sk* (*Figure 4A*, *Figure 4—figure supplement 4*). This inversion has been previously described (*Brown et al., 2011*; *Teresa Avelar et al., 2013*) and it occurred in the *Sp* lineage (*Brown et al., 2011*). Odd numbers of crossovers within this inversion would cause lethal chromosomal rearrangements (duplications of one arm and deletion of the other). This would cause spore inviability and likely explains why *rec12⁺ Sk/Sp* hybrids do not have higher fertility than *rec12Δ* hybrids, since odd numbers of crossovers within the inversion would likely impair viable spore recovery as much as an absence of recombination (*Figure 3*). The recombination pattern of markers across chromosome 1 is consistent with this interpretation. The *lys1-lys7* interval, in which we observed the greatest reduction in recombinants, is located within the inversion (*Figure 4A*, *Figure 4—figure supplements 2, 3*). The next highest reduction in recombinant frequencies was between markers flanking the inversion boundary (*Figure 4A*, *Figure 4—figure supplements 2, 3*). In contrast, recombination frequencies outside the inversion were only ~twofold to eightfold decreased in hybrids compared *Sp/Sp* diploids (*Figure 4A*, *Figure 4—figure supplements 2, 3*). These non-inversion associated differences are not a hybrid-specific defect and may be due to different recombination frequencies in *Sk* and *Sp*, perhaps caused by different DSB frequencies or different DSB repair outcomes (*Figure 4—figure supplement 1*).

Our analyses of recombination also revealed a surprising genetic linkage between *leu1* and *ade6* in *Sk/Sp* hybrids, despite the fact that these genes are located on chromosomes 2 and 3, respectively, in both species (*Figure 4B*, *Figure 4—figure supplement 2*). We speculated that there was a reciprocal translocation between *Sk* chromosomes 2 and 3, relative to *Sp*. If such a

*Figure 2. Continued*

apparent at the same sites in *Sk/Sp* and *Sk/Sk*. (**E**) DSBs are efficiently repaired in *Sk/Sp*. Ethidium bromide stained pulsed-field gel of *rad50⁺* diploids at the given times after the induction of meiosis show that DSBs do not accumulate more in *Sk/Sp* than the *Sp/Sp* control during meiotic prophase. Together with those in (**B**) these data demonstrate that *Sk/Sp* cells form and efficiently repair DSBs.

The following figure supplements are available for figure 2:

**Figure supplement 1**. DSB hotspots in *Sk* and *Sp*. DOI: 10.7554/eLife.02630.005

**Figure supplement 2**. DSB hotspots in *Sk* and *Sp*. DOI: 10.7554/eLife.02630.006

**Figure supplement 3**. DSB hotspots in *Sk* and *Sp*. DOI: 10.7554/eLife.02630.007

**Figure supplement 4**. DSB hotspots in *Sk* and *Sp*. DOI: 10.7554/eLife.02630.008

**Figure supplement 5**. DSB hotspots in *Sk* and *Sp*. DOI: 10.7554/eLife.02630.009

**Figure supplement 6**. DSB hotspots in *Sk* and *Sp*. DOI: 10.7554/eLife.02630.010

**Figure supplement 7**. DSB hotspots in *Sk* and *Sp*. DOI: 10.7554/eLife.02630.011

**Figure supplement 8**. DSB hotspots in *Sk* and *Sp*. DOI: 10.7554/eLife.02630.012

**Figure supplement 9**. DSB hotspots in *Sk* and *Sp*. DOI: 10.7554/eLife.02630.013

**Figure supplement 10**. DSB hotspots in *Sk* and *Sp*. DOI: 10.7554/eLife.02630.014

**Figure supplement 11**. DSB hotspots in *Sk* and *Sp*. DOI: 10.7554/eLife.02630.015

translocation included essential genes, it would render gametes with non-parental chromosome combinations of the affected arms inviable; this inviability would also create the semblance of genetic linkage between the two chromosomes. Consistent with this possibility, we found using Southern blot analyses that essential genes (*alr2* and *SPCP1E11.08*) had swapped chromosome locations between *Sk* and *Sp* (*Figure 4B,C*; *Kim et al., 2010*). We partially assembled the *Sk* genome to map the translocation junctions to position 676,281 on *Sp* chromosome 2 and 1,932,034 on *Sp* chromosome 3. We verified the translocation junctions via PCR (*Figure 4—figure supplement 5*). Analyses of synteny in two outgroup species (*S. octosporus* and *S. cryophilus*; *Rhind et al., 2011*) showed that the translocation occurred in the *Sk* lineage. The translocation appears to have resulted from a crossover between a Tf transposon found in *Sk* on chromosome 2 (corresponding to a single Tf transposon LTR in *Sp*) and a Tf transposon unique to *Sk* on chromosome 3.

Thus, we find that chromosomal rearrangements are likely a significant contributor to *Sp/Sk* hybrid infertility. However, our findings suggested that recombination defects due to chromosome rearrangements are not sufficient to explain the near complete hybrid sterility we see in *Sk/Sp* hybrids for several reasons. First, *rec12Δ Sk/Sp* hybrid diploids still have greater than six-fold lower viable spore yield than *rec12Δ Sk/Sk* or *Sp/Sp* (pure species) diploids (*Figure 5A*). The chromosome 2-chromosome 3 reciprocal translocation predicts only a two-fold lower viable spore yield because only half of the gametes would inherit an incompatible chromosome combination. Second, if defects in recombination cause errors in chromosome segregation leading to the production of aneuploid and diploid gametes (e.g., *Davis and Smith, 2003*), we would expect that the frequency of aneuploid and diploid spores produced by *Sk/Sp* hybrids in the absence of meiotic recombination (*rec12Δ*) should be similar to the frequencies observed in *rec12Δ* pure species controls. In contrast to this expectation, we find that the level of heterozygous aneuploids and diploids in *rec12Δ* hybrids is still significantly (at least twofold) higher than that observed in both *rec12Δ Sk/Sk* and *rec12Δ Sp/Sp* controls (*Figure 5A,B*; p<0.01 for both; G-test). This result indicates that something other than recombination defects contributes to the high fraction of diploids and aneuploids, as well as the low viable spore yield.

There were two possibilities to explain the higher recovery of aneuploid or diploid spores generated by *Sk/Sp* hybrids. First is the possibility that aneuploidy or diploid spores arise more frequently in hybrid meiosis. In contrast, the second possibility is that aneuploids arise at the same frequency in hybrid meiosis but are more likely than haploids to survive hybrid meiosis; haploid gametes produced by *Sk/Sp* hybrids may die more frequently than aneuploids. To distinguish between these possibilities, we calculated the frequency of aneuploids and diploids produced *per meiosis* in hybrids and pure species controls by multiplying the viable spore yield (viable spores produced per cell induced to undergo meiosis) by the fraction of viable spores that were aneuploid or diploid (i.e., aneuploids or diploids per viable spore). Our analyses revealed very small differences in the proportion of aneuploids that were generated in hybrids vs pure species meiosis (*Figure 5A*). We therefore conclude that the higher recovery of aneuploid and diploid spores from *Sk/Sp* hybrids is a result of lower viability of haploid spores.

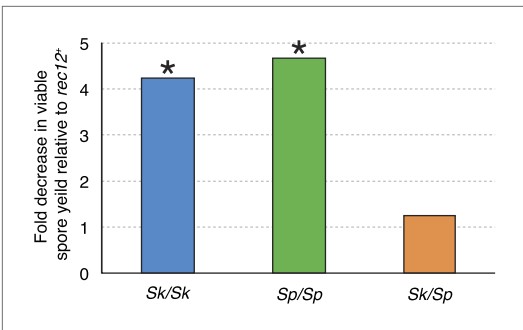

**Figure 3**. Recombination does not alter *Sk/Sp* hybrid fertility. The average *rec12*+ viable spore yield of each diploid was divided by that of the corresponding *rec12Δ* mutant. For the pure species diploids, the viable spore yield was significantly lower in the absence of Rec12 (n ≥ 5 experiments for each genotype; * *t* test p<0.05). The viable spore yield of *Sk/Sp* hybrids, on the other hand, was not significantly different between *rec12*+ and *rec12Δ Sk/Sp* hybrids (p=0.42). This indicates that recombination likely hurts fertility just as much as it promotes fertility in *Sk/Sp* hybrid meiosis. These data are shown in a different format in *Figure 5A*.

The lower viability of haploid spores could be partially explained by the incompatibility between *Sp* and *Sk* chromosomes 2 and 3 due to the reciprocal translocation in *Sk*. Aneuploids with *Sk* and *Sp* chromosome 3 (chromosome 3 disomes) will contain all essential genes, whereas haploids must inherit chromosomes 2 and 3 from the same species to be viable. This scenario, however, predicts only a twofold advantage of aneuploidy, yet *rec12Δ Sk/Sp* hybrids produce at least threefold more aneuploids than pure species *rec12Δ* controls. Furthermore, eliminating the chr2-chr3 incompatibility (see below) did not mitigate the phenotype. We therefore hypothesize that the viability benefit conferred by aneuploidy is due to disomy or heterozygosity for an allele(s) on chromosome 3. This could, for example, be due to a protective effect exerted by the *Sk* chromosome 3 within gametes containing *Sp* chromosome 3 and *vice versa* ('Discussion').

## Non-Mendelian transmission of alleles through hybrid meiosis

One explanation for a chromosome rearrangement-independent mechanism of hybrid infertility emerged from our studies following the transmission of alleles from each species' chromosome through meiosis in *Sk/Sp* hybrids. To focus on cells with only one copy of a given allele, we omitted heterozygous diploids from analyses of all alleles and omitted heterozygous aneuploids when analyzing transmission of alleles on chromosome 3. Surprisingly, we found that in all cases, the *Sk* allele was transmitted to more viable spores than the *Sp* allele (*Figure 6A,B*, *Figure 6—figure supplement 1*). This phenotype was not specific to the markers used, or which species contributed the marked (mutant) allele to the *Sk/Sp* hybrid diploid. In addition, the overrepresentation of *Sk* alleles from chromosomes 1 and 2 in the viable spores was observed within both haploid and aneuploid spores (*Figure 6—figure supplement 2*). This suggests the enhanced transmission of *Sk* alleles on these chromosomes is independent of the aneuploidy phenotype.

Since yeast meiosis is symmetric (i.e., all four meiotic products can be successfully packaged as gametes), differential transmission of alleles indicates that a nonrandom subset of the meiotic products must be rendered inviable. Given that we observe over-representation of *Sk* alleles in the viable spores of *Sk/Sp* hybrids, this implies that there must be a corresponding death of spores inheriting *Sp* alleles. This nonrandom death of gametes is therefore an additional cause of *Sk/Sp* hybrid infertility. For ease of reference, we refer to this phenotype as 'drive' of *Sk* alleles although this is not meiotic drive as originally envisioned to occur in asymmetric (female) meiosis (*Sandler and Novitski, 1957*). The term has, however, long been accepted to describe genetic elements that bias allele transmission by acting after the meiotic divisions (*Zimmering et al., 1970*).

Genetic markers nearer to any driving locus would be expected to show a stronger transmission bias than those that are more distantly linked. This allowed us to coarsely map the locations of the *Sk* driving loci using data from our hybrid recombination analyses (*Figures 6A*, *Figure 4—figure supplement 3*). On chromosome 1, the transmission bias of *Sk lys1* (72%) was stronger than those of both *ura1* (55%) and *arg3* (58%), indicating that the driving locus is closer to *lys1* than to *ura1* or *arg3*. On chromosome 2, the driving locus is closer to *leu1* (94% *Sk* transmission bias) than to *his5* (67%), *his4*, *lys4*, or *ade8* (all ~64%). The driving locus on chromosome 3 is closer to *ade6* (82%) than to *ura4* (52%). The significant overrepresentation of *Sk* alleles on all three chromosomes was also observed in spores produced by *rec12Δ Sk/Sp* hybrids (p<0.01 G-test; *Figure 6B*, *Figure 6—figure supplement 1*). However, the transmission of alleles was altered by the absence of recombination. This was expected due to stronger linkage with the driving locus in the absence of recombination. For example, *his5* is distantly linked to the chromosome 2 driving locus in *rec12*+ but appears more closely linked

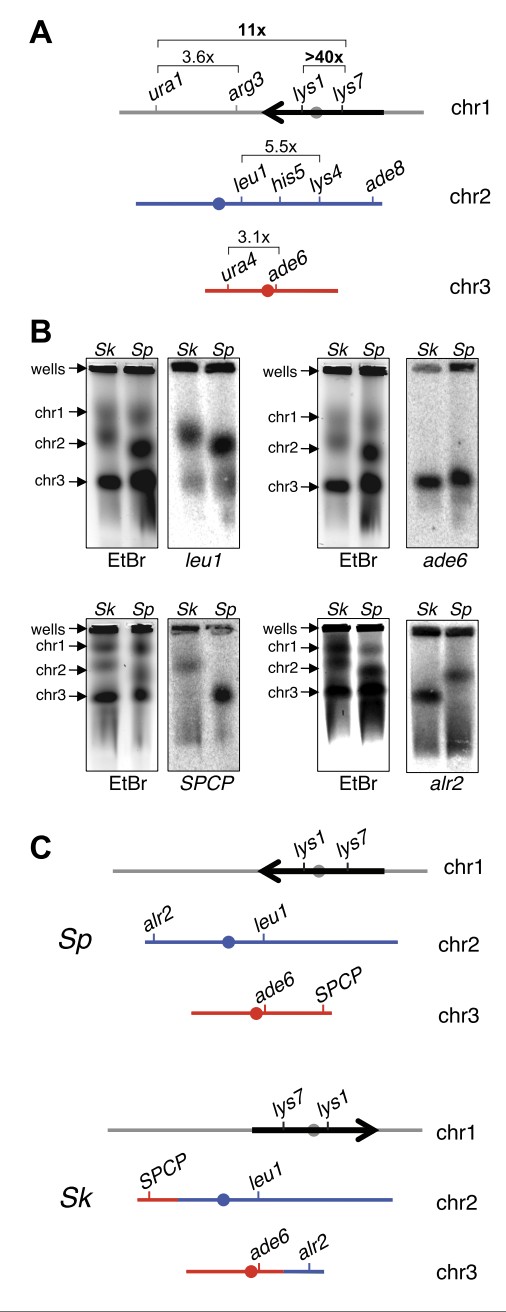

**Figure 4**. Chromosome rearrangements limit the recovery of recombinant progeny in *Sk/Sp* hybrids. (**A**) A cartoon illustrating the fold decrease in recombinant frequencies in *Sk/Sp* spores compared to that in *Sp/Sp* spores. The detailed recombination data are in *Figure 4—figure supplements 2 and 3*. The *Sp* karyotype is depicted with a grey/black chromosome 1, blue chromosome 2, and red chromosome 3. The backwards black arrow indicates an inversion in *Sp* relative to the ancestral karyotype. (**B**) Pulsed-field gels separating *Sp* and *Sk* chromosomes and Southern blots of the gels probed with DNA from the indicated loci revealed a reciprocal translocation that includes several
*Figure 4. Continued on next page*

to it in the absence of recombination in *rec12Δ* (*Figure 6B*).

## Genic incompatibilities fail to explain drive of *Sk* alleles on all chromosomes

We considered the alternate possibility that the overrepresentation of *Sk* alleles might not reflect drive but could instead reflect genic incompatibilities. For example, if *Sp* gene A was incompatible with *Sk* gene B, but *Sk* gene A was compatible with both *Sp* and *Sk* alleles of gene B, *Sk* gene A would be over-represented in the surviving gametes. This scenario predicts that we still should expect to recover haploid spores with all three *Sk* chromosomes at the same frequency as haploids with all three *Sp* chromosomes, since pure species chromosome combinations are compatible. However, *rec12Δ Sk/Sp* meiosis yielded the pure *Sk* haploid genotype approximately five times more frequently (6.5% of viable spores) than the pure *Sp* haploid genotype (1.4%; *Figure 6—figure supplement 3*). In addition, we found that all haploid combinations of *Sk* and *Sp* chromosomes (except those with non-parental combinations of chromosomes 2 and 3) are viable and have similar growth rates, suggesting that they do not suffer from a mitotic fitness loss. For example, cells that inherit *Sp* chromosome 1 (P1) and *Sk* chromosomes 2 and 3 (K2 and K3) grow very similarly to *Sp* cells. The reciprocal is also true (*Figure 7*). Although we cannot generate K2 P3 or P2 K3 haploids, rare recombinant strains that combine portions of K2 with portions of P3 (and vice versa) also grow well. Despite this, chromosome combinations containing *Sp* chromosomes are dramatically underrepresented amongst the viable spores of *rec12Δ Sk/Sp* hybrids (*Figure 6—figure supplement 3*). The overrepresentation of *Sk* alleles amongst the viable progeny therefore cannot be explained by (BDM) incompatibilities between nuclear *Sk* and *Sp* genes. Instead, our results are consistent with the action of meiotic drive alleles on *Sk* chromosomes that specifically target *Sp* chromosomes during or following hybrid meiosis. We have not, however, eliminated the formal possibility that genic incompatibilities exist and contribute to spore death in *rec12*[+] crosses, which do undergo meiotic recombination potentially exposing intrachromosomal incompatibilities.

Mitochondrial-nuclear incompatibilities are one cause of hybrid sterility in budding yeasts (*Lee et al., 2008*). Unlike in budding yeast, functional mitochondrial DNA (mtDNA) is essential for growth in wild-type fission yeast (*Wolf and Del Giudice, 1980*). Because of this,

*Figure 4. Continued*

essential genes including *alr2* and *SPCP1E11.08* (abbreviated *SPCP*). The EtBr-stained gels are on the left and the Southern blots are on the right in each pair. (**C**) A cartoon summary of the karyotype differences between *Sp* and *Sk*. The arrow indicates the location of the inversion in *Sp*. A few landmark loci are shown.

The following figure supplements are available for figure 4:

**Figure supplement 1**. Recombination frequencies in *Sk*.

**Figure supplement 2**. Recombination frequencies in *Sk*/Sp hybrids are low relative to *Sp*.

**Figure supplement 3**. *Sk* alleles are underrepresented in the progeny of *Sk/Sp* hybrids.

**Figure supplement 4**. *Sp* has an inversion on chromosome 1 relative to *Sk*.

**Figure supplement 5**. *Sk* has a reciprocal translocation between chromosomes 2 and 3.

mitochondrial-nuclear gene incompatibilities could be lethal. We therefore tested the hypothesis that incompatibilities between the *Sk* mitochondrial genome and *Sp* nuclear genes could also contribute to the meiotic drive phenotype. We did this by assaying the transmission of alleles through hybrids containing either *Sp* or *Sk* mtDNA (obtained after prolonged growth of *Sk/Sp* hybrid diploids). We genotyped the mtDNA using two distantly spaced single-nucleotide polymorphisms that alter restriction sites. We found that drive of *Sk* alleles was not significantly different between *rec12Δ* hybrid diploids with *Sk* or *Sp* mtDNA (G-test; *Figure 6C*, *Figure 6—figure supplement 4*). This indicates that *Sk* drive is not caused by an incompatibility between mitochondrial and nuclear genomes.

## Each *Sk* chromosome contains a driving allele

Given the observed reciprocal incompatibility between *Sk* and *Sp* chromosomes 2 and 3 due to the translocation, we reasoned that the observed drive phenotypes of *Sk* alleles on these chromosomes could be interdependent. This scenario is easiest to understand in the *rec12Δ* crosses, which are not complicated by recombination. For example, if allele(s) on K2 drive, K3 may appear to drive just because spores that inherit K2 and P3 are inviable. The converse could also be true. This line of thinking inspired us to test whether each *Sk* chromosome can drive independently, in the absence of drive on the other two chromosomes. We used *rec12Δ* strains to eliminate recombination-associated phenotypes. To test for the ability of K1 to drive autonomously, we crossed a P1 K2 K3 strain (obtained from a *rec12Δ* hybrid cross) to a naïve K1 K2 K3 strain to generate a P1 K2 K3/K1 K2 K3 diploid. We found that even in these diploids, which were heterozygous for only chromosome 1, the K1 chromosome showed the drive phenotype

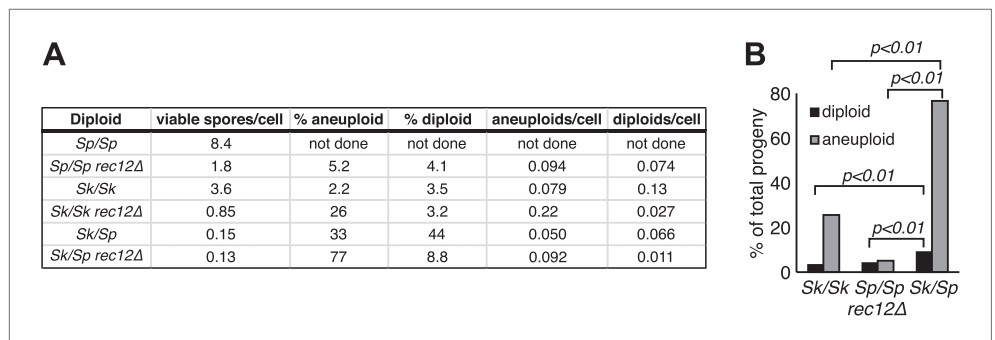

**A**

| Diploid | viable spores/cell | % aneuploid | % diploid | aneuploids/cell | diploids/cell |
|---|---|---|---|---|---|
| *Sp/Sp* | 8.4 | not done | not done | not done | not done |
| *Sp/Sp rec12Δ* | 1.8 | 5.2 | 4.1 | 0.094 | 0.074 |
| *Sk/Sk* | 3.6 | 2.2 | 3.5 | 0.079 | 0.13 |
| *Sk/Sk rec12Δ* | 0.85 | 26 | 3.2 | 0.22 | 0.027 |
| *Sk/Sp* | 0.15 | 33 | 44 | 0.050 | 0.066 |
| *Sk/Sp rec12Δ* | 0.13 | 77 | 8.8 | 0.092 | 0.011 |

**Figure 5**. Increased aneuploidy amongst viable *Sk/Sp* gametes is recombination-independent. (**A**) We calculated both viable spore yield (viable spores/cell) as well as the fraction of viable spores that are aneuploid or diploid ('Materials and methods'). In the absence of Rec12, the relative frequencies of aneuploids and diploids are elevated in all cases. However, there is significantly more aneuploidy and diploidy of viable spores produced by *rec12Δ Sk/Sp* hybrids than by *rec12Δ* pure species controls. This shows the phenotype is not caused solely by recombination defects. In addition, *Sk/Sp* diploids do not generate more aneuploids or diploids relative to the number of cells induced to undergo meiosis compared to pure-species controls. Some of these data are presented in a different format in *Figures 1H* and *Figure 3*. (**B**) A bar graph illustrating the fraction of the viable spores produced by the indicated *rec12Δ* diploids that are aneuploid or diploid (G-test, n >300 for each).

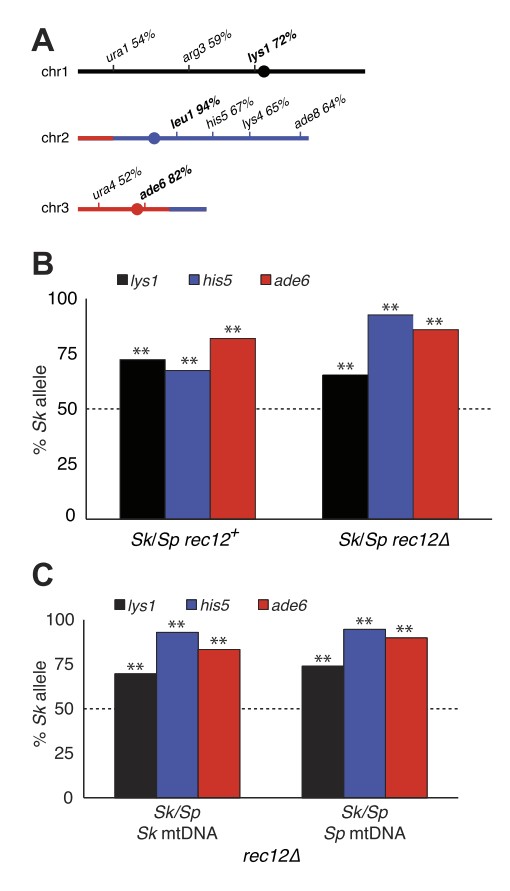

**Figure 6**. Alleles on all three *Sk* chromosomes show drive (independent of mitochondrial DNA type). (**A**) *Sk* alleles were inherited by significantly more than 50% of the viable spores produced by *Sk/Sp* hybrids, except *ura1* and *ura4* (G-test p<0.01; n >100 for each). The markers nearest to the meiotic drive loci (i.e., those showing the greatest bias towards *Sk* inheritance) are shown in boldface. The color scheme is the same as that in *Figure 4*. The data underlying these numbers are shown in *Figure 4—figure supplement 3*, and *Figure 6—figure supplement 1*. (**B**) The *Sk* alleles of *lys1*, *his5* and *ade6* show significant drive both in the presence and absence of recombination (**p<0.01, n >300 for *lys1* and *his5*, n >80 for *ade6*). The amount of *his5* drive is greater in the absence to Rec12 due to enhanced linkage with the driving locus. The data underlying this graph are shown in *Figure 6—figure supplement 1*. (**C**) Incompatibilities between the *Sk* mitochondrial DNA and *Sp* nuclear genes are not responsible for the drive phenotype because we observed the same drive in *rec12Δ Sk/Sp* hybrids with either *Sk* or *Sp*-derived mitochondrial DNA (**p<0.01, n >200 for *lys1* and *his5*, n >50 for *ade6*). The data underlying this graph are shown in *Figure 6—figure supplement 4*.

*Figure 6. Continued on next page*

(G-test p<0.01 *Figure 8A*, *Figure 8—figure supplement 1*), proving that the driving allele on K1 can act even in the absence of K2 and K3 drive.

Testing the ability of K2 and K3 to drive autonomously was complicated by the reciprocal translocation, as we could not construct K1 P2 K3 and K1 K2 P3 strains because they would lack some essential genes. To get around this, we took advantage of the fact that rare Rec12-independent recombination can occur during mitosis or meiosis, for example at the mating type locus on chromosome 2 (*Davis and Smith, 2003*; *Angehrn and Gutz, 1968*). Because of this rare Rec12-independent recombination, we recovered some viable spores (<1%) from *rec12Δ Sk/Sp* hybrid meioses that appeared to have non-parental combinations of chromosomes 2 and 3, based on the phenotypic markers we used. Upon further examination via PCR or whole genome sequencing, these all proved to be rare recombinants that contain the left end of chromosome 2 and the right end of chromosome 3 from the same parent. We fully sequenced two of these recombinant strains to map their recombination breakpoints (*Figure 8—figure supplement 1*). These two strains both contained recombinant chromosomes that have the *Sk* karyotype, but showed the *Sp* phenotype in crosses to *Sk* (i.e., they were underrepresented in the viable gametes). We denote these as 'R' (for 'recombinant') chromosomes. By crossing strains containing these 'R' chromosomes to naïve *Sk*, we found that K2 and K3 chromosomes are also able to drive autonomously (*Figure 8A*, *Figure 8—figure supplement 2*). Specifically, in the K1 R2 K3/K1 K2 K3 diploid, only K2 showed meiotic drive. Similarly, only K3 exhibited drive in the K1 K2 R3/K1 K2 K3 diploid.

Our experiments showed that both K2 and K3 have the ability to drive autonomously (*Figure 8A*). Intriguingly, these diploids also revealed that K2 meiotic drive was less intense in the absence of linked K3 drive (61% instead of 93%; p<0.01). In contrast, K3 drives more strongly in the absence of a linked driving K2 (91% vs 86%; p=0.02). These changes in drive intensity could be affected by genetic interactions between chromosomes revealed by the specific genotypes of these strains (*Figure 8—figure supplement 1*). However, they could also be due to removing the reciprocal translocation. In pure *Sk/Sp* diploids, gametes that inherit *Sk* chromosome 3 would also have to inherit *Sk* chromosome 2 to survive. In this way, the driving ability of a weaker meiotic drive allele (e.g., on K2) could be augmented via pseudo-linkage to a stronger drive allele (on K3).

*Figure 6. Continued*

The following figure supplements are available for figure 6:

**Figure supplement 1**. Summary of *Sk/Sp* hybrid and pure species diploid meiotic phenotypes and distribution of alleles in their progeny.

**Figure supplement 2**. Biased transmission favoring *Sk* alleles on chromosomes 1 and 2 is observed in aneuploid and haploid spores.

**Figure supplement 3**. Distribution of progeny from *rec12Δ Sk/Sp* hybrid meiosis.

**Figure supplement 4**. Meiotic drive in *Sk/Sp* hybrids is independent of mitochondrial DNA.

If the drive of *Sk* alleles we observe in *Sk/Sp* hybrids is caused by nonrandom death of spores that inherit *Sp* alleles, then the magnitude of drive we observe should correlate with level of infertility we observe. To test this, we calculated viable spore yields for a series of diploids with varying amounts of meiotic drive (*Figure 8A*). Consistent with our hypothesis, we observed that the fertility of diploids was correlated with the magnitude of drive. For example, the P1 K2 K3/K1 K2 K3 and the K1 R2 K3/K1 K2 K3 diploids had the lowest amount of drive and the highest viable spore yield (*Figure 8A,B*, *Figure 8—figure supplement 1*). In contrast, the K1 K2 R3/K1 K2 K3 diploid had the most drive and the lowest viable spore yield. These results are consistent with nonrandom death of gametes being directly associated with the over-representation of *Sk* alleles.

These diploids containing one heterozygous chromosome also fortuitously allowed us to home in on the cause of the high frequency of aneuploids we observed amongst the viable spores of *Sk/Sp* hybrids. We predicted the phenotype was due to heterozygosity for *Sk* and *Sp* alleles on one or more chromosomes, because the phenotype is specific to *Sk/Sp* hybrids. Amongst the viable spores produced by both the P1 K2 K3/K1 K2 K3 and K1 R2 K3/K1 K2 K3 diploids, the frequency of aneuploid spores was similar to that of the pure species *rec12Δ Sk/Sk* control (≤30%; *Figure 8C*). This result further supports that the drive phenotype of *Sk* chromosomes 1 and 2 is independent of the aneuploidy phenotype of *Sk/Sp* hybrids, but it also shows that heterozygosity of these chromosomes does not cause the aneuploidy phenotype. Conversely, a much higher fraction (68%) of the viable spores produced by the K1 K2 R3/K1 K2 K3 diploid, which is heterozygous for most of chromosome 3, were aneuploid (*Figure 8—figure supplement 1*). The percentage of the viable spores from this diploid that were aneuploid was only marginally less than the value we observed in the spores produced by the completely heterozygous *Sk/Sp* hybrid *rec12Δ* diploids (78%; p=0.038; *Figure 8C*, *Figure 8—figure supplement 1*). These results show that the high frequency of aneuploids we observe amongst the viable spores of *Sk/Sp* hybrids is largely due to heterozygosity for a locus (or loci) on chromosome 3.

## Discussion

### Chromosomal rearrangements are frequent and significantly contribute to hybrid infertility in fission yeast

Our study reveals that the largest contributors to *Sk/Sp* hybrid infertility are two chromosomal rearrangements involving all three chromosomes. First, *Sp* chromosome 1 contains a large (~2.2 Mb, or >350 cM in *Sp*; about half of the chromosome) inversion relative to *Sk*, which retains the ancestral state (*Figure 4*). This inversion has been described before and is found in several fission yeast isolates in addition to the common *Sp* L972 lab strain used in our study (*Brown et al., 2011*; *Teresa Avelar et al., 2013*). This inversion contributes to hybrid infertility by limiting the possibility for interhomolog crossovers. Odd numbers of crossovers within this region would generate incomplete genomes and hence inviable gametes. There may be additional, smaller inversions present between these two species, which will be elucidated only by complete assembly of the *Sk* genome.

Second, we discovered a reciprocal translocation that occurred in the *Sk* lineage in which DNA that is found on the left end of P2 is found on K3, and DNA that is found on the right end of P3 is found on K2 (*Figure 4*). This translocation contributes to *Sk/Sp* hybrid infertility because each translocated sequence includes essential genes, rendering K2 P3 and P2 K3 chromosome combinations inviable. Although this translocation has been observed only in *Sk* thus far, translocations are remarkably common amongst fission yeast isolates closely related to *Sp* (*Brown et al., 2011*; *Teresa Avelar et al., 2013*).

Despite very little DNA sequence divergence (~0.5%) and little phenotypic divergence, there have been 12 translocations identified in fission yeast isolates closely related to *Sp* (*Brown et al., 2011*; *Rhind et al., 2011*; *Teresa Avelar et al., 2013*). These isolates have all been termed *S. pombe* for their

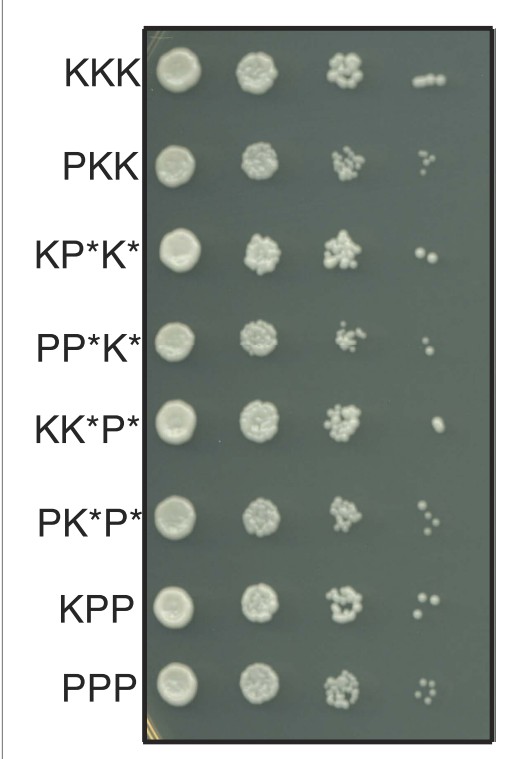

**Figure 7**. The haploid progeny of *Sk/Sp* hybrids have similar growth rates. The progeny of *rec12Δ Sk/Sp* hybrids with the indicated chromosomes were diluted and grown on rich YEA medium. KKK indicates the *Sk* parental genotype, whereas PPP indicates the *Sp* parental genotype. The strains were genotyped using *lys1, his5* and *ade6* alleles on chromosomes 1, 2, and 3, respectively. Strains that inherit intact chromosomes 2 and 3 from different species are non-viable because they lack many essential genes. However, we do rarely recover viable recombinant strains that have alleles from *Sk* chromosome 2 and *Sp* chromosome 3 (and vice versa). Potentially recombinant chromosomes are denoted with an *. All haploid strains recovered have growth rates similar to that of the parental species, suggesting mitotic growth defects do not underlie differential recovery of the genotypes.

phenotypic similarities and the low nucleotide divergence between them, despite the fact that many of these isolates might be significantly reproductively isolated from *Sp*, just like *S. kambucha* (**Brown et al., 2011**; **Teresa Avelar et al., 2013**). Like the *Sk/Sp* hybrids assayed in this work, genomic rearrangements are likely the largest contributors to hybrid infertility within other closely related fission yeasts. This high number of chromosomal rearrangements is remarkable, especially when compared to the *Saccharomyces* '*sensu stricto*' group of budding yeast species, in which only 10 translocations have been identified despite 50-fold higher DNA sequence divergence (**Fischer et al., 2000**).

## Meiotic drive alleles significantly (and separately) contribute to hybrid infertility

Two types of loci can achieve transmission distortion in their own favor during meiosis. The first are 'true drive' alleles in which the drive occurs due to nonrandom segregation of alleles during the meiotic divisions in asymmetric (female) meiosis. For example, 'knobs' on maize chromosomes can drive by preferentially segregating into the nucleus that will become the oocyte, rather than into nuclei that become polar bodies and are lost (**Dawe and Cande, 1996**). Similarly, centromeres in *Mimulus* species can drive in female meiosis (**Fishman and Saunders, 2008**). Alternatively, transmission distortion can be achieved not by nonrandom segregation but instead by 'gamete killing' in symmetric (male) meiosis. Gamete killer alleles segregate into gametes randomly, but the trans-acting killer alleles cause the death or malfunction of the gametes that do not inherit them. Although the mechanisms vary, this type of drive occurs in widely diverged eukaryotes. For example, male mice heterozygous for the t-haplotype produce mostly (>95%) functional sperm containing the t-haplotype. The sperm that do not inherit the t-haplotype are nonfunctional due to immobility (**Schimenti, 2000**). Similarly, in *Drosophila*, sperm with chromosomes carrying large amounts of the *Responder* DNA satellite suffer chromosome condensation delays in the presence of the *Segregation Distorter* haplotype and do not contribute equally to functional sperm (**Larracuente and Presgraves, 2012**). In *Neurospora*, a spore killer allele kills gametes that do not inherit it by interfering with spore development (**Hammond et al., 2012**).

We discovered meiotic drive loci on each of the three *Sk* chromosomes (**Figures 6, 8**). We do not yet know the mechanism by which *Sk* alleles gain their transmission advantage or at what point in gametogenesis it occurs. Drive likely occurs, however, via trans-acting gamete killing, as observed in all other cases of drive in symmetric meiosis, not via nonrandom chromosome segregation. We hypothesize that a 'poison' produced by each driving allele during hybrid sporulation renders spores that inherit the *Sp* allele at that locus less viable. For instance, such a 'poison' may prevent hybrid spores that inherit the *Sp* allele from expressing an essential gene for completing sporulation or

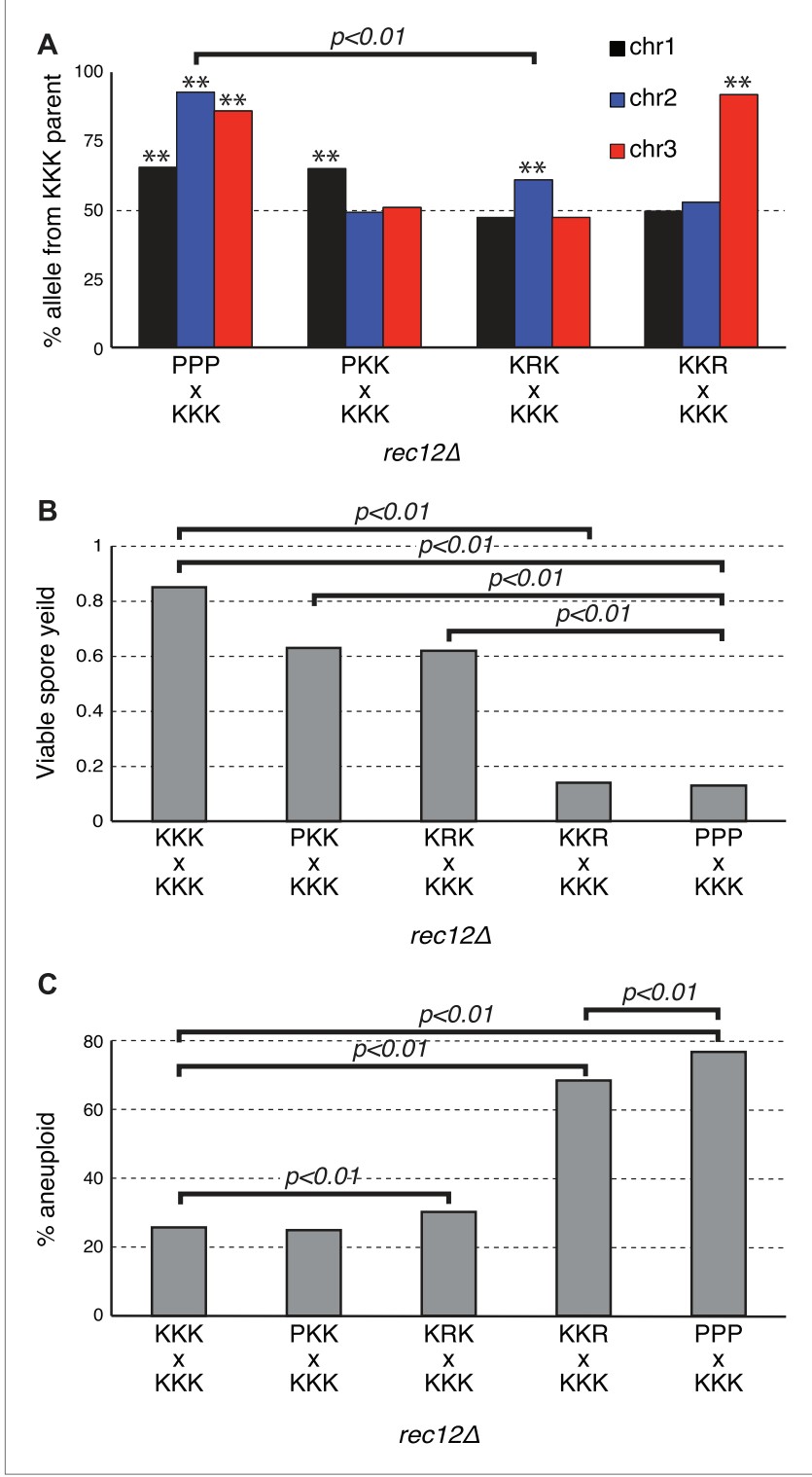

**Figure 8**. *Sk* drive alleles are autonomous and contribute to hybrid infertility. Aneuploidy is largely caused by heterozygosity of *Sk* and *Sp* DNA on chromosome 3. (**A**) Comparison of meiotic drive phenotypes between *rec12Δ* diploids generated by mating *Sk* to *Sp* or to haploid strains obtained from *Sk/Sp* hybrids. 'R' indicates a recombinant chromosome (***Figure 8—figure supplement 1***), which is compatible with all *Sk* chromosomes but does not contain a meiotic drive allele. All *Sk* chromosomes can drive autonomously (** indicates drive; G-test p<0.01). However, the drive of *Sk* chr2 is lower in the KRK/KKK diploid than in pure PPP/KKK hybrids (G-test; n >500 for chromosomes 1 and 2, n >80 for chromosome 3 in each cross). The PPP/KKK data are also shown in ***Figure 6B***.
*Figure 8. Continued on next page*

*Figure 8. Continued*

(**B**) Fertility defects of hybrids parallel the amount of drive observed amongst the viable spores (see A, p-values obtained from t-tests, averages of at least five experiments are shown). This is consistent with drive causing spore death. (**C**) The high aneuploidy amongst the viable progeny of *Sk/Sp* hybrids is largely due to heterozygosity of one or more loci on chromosome 3 (G-test, n >500 for each cross). The PPP/KKK and KKK/KKK viable spore yield and aneuploid data are also shown in *Figure 5*. The data underlying these graphs are summarized in *Figure 8—figure supplement 2*.

The following figure supplements are available for figure 8:

**Figure supplement 1**. Genotype of recombinant strains used in *Figure 8*.

**Figure supplement 2**. Summary of meiotic phenotypes for *Sk/Sp* hybrids and diploids with one heterozygous chromosome and the distribution of alleles in their viable progeny.

germination. Precise mapping of the driving alleles will help reveal their mechanisms of action. Genetically separating the action of each driver (*Figure 8*) allows us to conclude that they would each meet the criteria for meiotic drive. However, simultaneous action of all three drivers exposed in the *Sk/Sp* cross, together with the chromosomal rearrangements, leads to a dramatic loss of viable spore yield and would curb the fixation of the driver alleles. This is why we hypothesize that each of the three drivers that favor the transmission of *Sk* alleles likely arose independently or sequentially along the *Sk* lineage.

Natural selection works best when heterozygous alleles are transmitted at Mendelian (1:1) frequencies through heterozygotes (*Crow, 1991*). In this way, the alleles compete on an equal playing field and spread through a population by promoting fitness. By subverting Mendelian transmission, meiotic drive alleles and linked loci can spread and even drive more fit alleles to extinction. The meiotic drive we observed could therefore also be detrimental to fitness indirectly by biasing allele transmission. In this scenario, natural selection would therefore favor the evolution of drive suppressors that are unlinked to the driving loci (*Crow, 1991*). Meiotic drive loci that could evade such suppression and re-establish their transmission advantage would then be favored. This genetic conflict between meiotic drive loci and their suppressors would set up an evolutionary arms race where both sides must innovate to try to gain an advantage. We predict that such an arms race is occurring within fission yeasts and we are able to observe drive only in the hybrids because the drive suppressors in *Sp* are incompatible with the driving alleles in *Sk*. Under this scenario, it is completely possible that in a cross between *Sp* and another *Sp*-like isolate, the *Sp* drive alleles may emerge victorious.

Our study of *Sk/Sp* hybrids also revealed a previously unanticipated feature of a meiotic drive system. The reciprocal translocation between *Sk* chromosomes 2 and 3 genetically links the two chromosomes. For example, *leu1* and *ade6* are genetically linked in *Sk/Sp* hybrids despite being on chromosomes 2 and 3, respectively (*Figure 4B*, *Figure 4—figure supplement 2*). *leu1* and *ade6* are also both closely linked to the drive alleles on their chromosomes; thus, the translocation also links the drive alleles on separate chromosomes (*Figure 6A*). We demonstrated that each of the drive alleles can act in the absence of drive of the other two chromosomes (*Figure 8*). In an exciting twist for a meiotic drive system, however, the drive of the weaker allele may be bolstered by the drive of the stronger allele. Such a two-chromosome, two-locus drive system could be a formidable selfish evolutionary force. Like an inversion, the translocation expands the region effectively linked to the driving loci, resulting in a larger proportion of the genome favoring drive enhancers and a smaller proportion favoring suppressors. In addition, if a suppressor of one of the drive alleles does arise in the population, the drive of the other allele can compensate and minimal suppression will actually occur. In other words, both drive alleles must be suppressed simultaneously to restore Mendelian allele transmission.

## Meiotic drive and aneuploidy

Finding independent drivers on all three *Sk* chromosomes still does not explain our findings of increased frequencies of aneuploid (and diploid) gametes amongst the viable spores of *Sk/Sp* hybrids compared to the frequencies observed in pure species diploids, even in the absence of recombination

(*Figure 5*). Our crosses (*Figure 8*, *Figure 8—figure supplement 2*) indicate that the aneuploidy phenotype is caused by *Sk/Sp* heterozygosity of an allele or alleles on chromosome 3. The aneuploidy phenotype seems paradoxical given that aneuploidy is generally deleterious (*Niwa et al., 2006*). Indeed, even the aneuploids produced by *Sk/Sp* hybrids show a slower growth phenotype (*Figure 1F*). One hypothesis that would explain this higher recovery of aneuploid and diploid spores is that the disjunction of chromosomes (especially the *Sk* and *Sp* third chromosomes) is compromised during hybrid meiosis. Contrary to this, we calculate that hybrid meioses do not generate aneuploid or diploid gametes more frequently (*Figure 5A*). Rather, we hypothesize that aneuploid and diploid spores produced by *Sk/Sp* hybrids are more likely than haploid spores to survive to produce a colony.

Fitness advantages for aneuploids have been observed under some conditions and certain genetic backgrounds have increased aneuploidy tolerance (*Ni et al., 2013*; *Pavelka et al., 2010*; *Yona et al., 2012*; *Torres et al., 2010*; *Tange et al., 2012*). For example, aneuploidy could be beneficial to fitness by increasing the dosage of certain genes (*Torres et al., 2007*). However, we observed more than sevenfold more K1 K2 K3/P3 aneuploids (289) than K1 K2 K3 (38) haploids amongst the viable spores produced by *rec12Δ Sk/Sp* hybrids. K1 K2 K3 haploids represent the pure species genotype (and *Sk/Sk* diploids have high fertility), so it is hard to envision how adding P3 will provide a more than a sevenfold fitness boost, especially given the growth defects of aneuploids.

We favor an alternative model in which the aneuploidy and meiotic drive phenotypes are interconnected. We propose that *Sk/Sp* aneuploids are more resistant to poisons produced by meiotic drive alleles acting in hybrids. Our model posits that the *Sp* third chromosome encodes a weaker driver (poison) that acts against *Sk* chromosome 3, whereas the stronger *Sk* driver (poison) acts against *Sp* chromosome 3. In this model, only cells containing both *Sk* and *Sp* chromosome 3 are fully resistant to the effects of both drivers. Because aneuploids are genetically unstable and frequently become haploid upon mitotic growth (*Niwa et al., 2006*; *Kohli et al., 1977*), the enhanced viability of aneuploid (and likely diploid) spores must be manifest during spore formation, spore germination, or the first few mitotic divisions after germination. Testing this model and other alternatives will require identification of the loci responsible for drive. Regardless of their mechanism(s) of action, the meiotic drive alleles we identified contribute significantly to hybrid spore death (*Figure 8C*). *Sk* alleles are preferentially found in the viable spores, whereas *Sp* alleles are presumably enriched amongst the dead spores.

## Multiple step model for speciation in fission yeasts

Why are rearrangements so common in fission yeasts? *Brown et al. (2011)*, who first discovered the karyotype diversity in different *S. pombe* isolates, suggested that these organisms might live in small populations and rarely outcross. Teresa Avelar et al. then proposed that rearrangements could spread due to fitness advantages provided by the rearrangement during mitotic growth (*Teresa Avelar et al., 2013*; *Colson et al., 2004*). In both these scenarios, a chromosomal rearrangement could spread to fixation within a population without ever undergoing meiosis in a heterozygous state. These rearrangements would then be similar to Dobzhansky-Muller incompatibilities in that the derived karyotypes evolved in separate populations and were never tested by selection. Subsequent meioses of heterozygotes comprised of different derived karyotypes would be impaired and lead to reproductive isolation, impeding gene flow between the two populations. In theory, having a sufficiently large number of chromosomal rearrangements should suffice to ensure complete sterility.

We find that chromosomal rearrangements alone are insufficient to explain near complete hybrid sterility in a case of incipient speciation in fission yeast. In addition we show that at least three drive loci exist in *Sk*. We suspect that meiotic drive alleles are common in fission yeasts. Meiotic drive alleles have been proposed to promote karyotype evolution in diverse eukaryotes, and it is possible that drive also played a role in the evolution of the reciprocal translocation we identified in *Sk* (*White, 1978*; *Pardo-Manuel de Villena and Sapienza, 2001*; *Dyer et al., 2007*). As we have pointed out, it is highly unlikely that the three meiotic drive alleles and the two chromosomal rearrangements we have found in the *Sk/Sp* hybrids occurred simultaneously during evolution, especially given the fertility costs. It is more likely they accumulated in stepwise fashion over time. The severe fertility phenotypes we observe in the current-day hybrids therefore do not entirely recreate the exact fitness costs and benefits of each novel mutation (rearrangements, drive) faced during their evolutionary histories, although they serve as effective proxies.

Our findings in fission yeast are highly reminiscent of emerging themes from studies of reproductive isolation mechanisms in plants, insects and even mammals, in which genetic conflicts underlie

hybrid incompatibilities (*Johnson, 2010*; *Presgraves, 2010*). Here we provide additional support for such models by showing that three genetic conflicts involving selfish meiotic drive alleles directly contribute to hybrid infertility in fission yeasts. Thus, like other sexually reproducing eukaryotes, genetic conflicts could also be fundamental drivers of infertility within fission yeasts. Our study finds important contributions from both chromosomal and genic divergence in mediating hybrid sterility in fission yeast, reminiscent of the chromosomal speciation model first proposed by *White (1978)*.

## Materials and methods

### Strain construction

We introduced all DNA transformation cassettes into the genome using a standard lithium acetate transformation protocol. All deletion strains were verified by PCR and phenotype analyses. All strains are listed in *Supplemental file 1*. The oligos used in strain construction are listed in *Supplemental file 2*. The *Sk* auxotrophic mutations were all complete deletions of open reading frames. For the auxotrophic mutations, we generated DNA deletion cassettes via PCR using 100 base pair (bp) oligos with 79 bp of homology on the 5′ ends to the target gene and 21 bp at the 3′ end homologous to DNA flanking drug resistance genes *kanMX6*, *natMX4*, and *hphMX4* (*Goldstein and McCusker, 1999*).

The *rec12Δ::ura4+ Sk* allele was constructed in a manner similar to that for the auxotrophic mutations. We used 100 bp oligos to amplify a *ura4+* DNA cassette with 79 bp 5′ and 3′ tails homologous to the DNA upstream and downstream, respectively, of the *rec12+* open reading frame. The transformation cassette included the entire *ura4+* coding sequence plus 525 bp of upstream sequence and 772 bp of downstream sequence.

To construct the *natMX4::rec12::His$_6$FLAG$_2$* allele, we first amplified the region from 927–431 bp upstream of the *rec12+* open reading frame with PCR oligos that added HindIII sites to either end of the PCR product. We then cloned the PCR product into the HindIII site of plasmid pAG25, which contains the *natMX4* gene (*Goldstein and McCusker, 1999*). We next amplified from pJF32 the *rec12:: His$_6$FLAG$_2$* allele plus 385 bp of DNA upstream and 173 bp of DNA downstream of *rec12* using PCR oligos that added a ClaI site to the 5′ end and an EcoRV site to the 3′ end (*Cromie et al., 2007*). We then cloned that PCR product downstream of the *natMX4* gene in the above (pAG25 derivative) plasmid to make a plasmid with 5′ homology to DNA upstream of *rec12*, followed by *natMX4*, then the *rec12:: His$_6$FLAG$_2$* allele. We then cut the final plasmid with NotI to release the DNA cassette for transformation. We verified the resulting transformants using PCR and DNA sequencing.

To construct the *rad50S* (i.e., *natMX4::rad50-K81I*) allele, we first used PCR to amplify the DNA from 394 bp upstream of *rad50* to 310 bp into the open reading frame (*Alani et al., 1990*). The oligos added a SmaI site 357 bp upstream of the *rad50* translational start site and added the K81I mutation (changing the codon from AAA to ATT). We cloned that PCR product into a pCR2.1 TOPO vector (Invitrogen, Grand Island, NY). We then cloned the PvuII-EcoRV fragment from pAG25 that included the *natMX4* gene into the SmaI site of the above plasmid (*Goldstein and McCusker, 1999*). We then amplified the entire *natMX4::rad50-K81I* allele from the plasmid using DNA oligos that added an additional 80 bp of DNA to the end of the transformation cassette to promote targeting of the DNA to the proper site in the genome (replacing the first part of the *rad50+* CDS with the cloned *rad50-K81I* mutant version). We verified the mutant strains via PCR, sequencing and phenotypic analyses.

### Isolation of recombinant strains

The recombinant strains used in *Figure 8*, shown as PKK (to indicate chromosome 1 from *Sp*, and chromosome 2 and chromosome 3 from *Sk*), KRK (R indicates a recombinant chromosome) and KKR, were all isolated from the spores produced by a *rec12Δ Sk/Sp* diploid in which there is very little meiotic recombination. These are strains SZY147, SZY240 and SZY302, respectively, in *Supplemental file 1*. We inferred the chromosome composition of these strains using *lys1*, *his5*, and *ade6* alleles and their meiotic phenotypes. The PKK strain contains the *Sp* allele of *lys1* (on chromosome 1) and the *Sk* alleles of *his5* (on chromosome 2) and *ade6* (on chromosome 3). The KRK strain contains the *Sp* allele of *his5* and the *Sk* alleles of *lys1* and *ade6*. The KKR strain inherited the *Sk* alleles of *lys1* and *his5*, but the *Sp* allele of *ade6*. However, in these last two strains, we reasoned that chromosome 2, chromosome 3 or both must be recombinant because the strains are viable (*Sp* and *Sk* chromosomes 2 and 3 are incompatible due to the translocation of essential genes). For clarity, we denote the third chromosome of KKR (and the second chromosome of KRK) as recombinant because they display the *Sp* meiotic phenotype (transmission to less than half of the spores)

when crossed to *Sk*, whereas the others display the *Sk* meiotic phenotype (transmission to half of the spores) when crossed to *Sk*. We reasoned the KRK and KKR must have the *Sk* karyotype because all chromosomes were compatible with naïve *Sk* chromosomes. To test our assumptions about where these stains' chromosomes were recombinant, we sequenced SZY302 and SZY240. The contributions of both *Sk* and *Sp* DNAs to these strains are summarized in *Figure 8—figure supplement 1*. The sequencing data for these hybrids are available in the NCBI sequence read archive (accession no. PRJNA245039).

### Induction of meiosis in liquid cultures and assaying the timing of the meiotic divisions

We used a protocol based on that of *Doll et al. (2008)*. We used single diploid colonies (see below) to start 10 ml YEL overnight cultures at 30°. The next day we diluted those cultures (1:100 or 1:50) into 100 ml PM medium and grew the cultures overnight (~16 hr) at 30° to an $OD_{600}$ ~1. We then pelleted the cells from 50 ml of culture and resuspended them in 500 ml of PM minus nitrogen medium to induce sporulation. The cells sporulated at 30°. To assay meiotic divisions, we sampled cells at the indicated time points, fixed them in formaldehyde and DAPI stained the cells as in *Malone et al. (2004)*. We then visualized the cells in a Zeiss Axiovert 200 M microscope.

### Fertility, meiotic drive, and recombination assays

To generate diploids, we first grew YEL cultures of each haploid parent overnight at 30°. We then mixed roughly equal volumes (<500 µl) of the cultures in microfuge tubes, pelleted the cells and spread them on SPA medium. We left the SPA plates at room temperature overnight (~16 hr) to allow the cells to mate. We then scooped up the mated cell mixture and spread it on minimal yeast nitrogen base (YNB) medium to select diploids. The haploid parents contained complementing auxotrophic markers, so the minimal medium selected diploids. After 2 days at 32°, we isolated single diploid colonies from the minimal plates and restreaked them to single colonies on fresh minimal medium. After 2 days at 32°, we used single diploid colonies to start small (3–4 ml) YEL cultures that we grew overnight at 30°. The next day, we plated a small volume (50–100 µl) of each culture on SPA to induce the cells to undergo meiosis at 25°. We also determined titers of each culture on YEA plates that we incubated 2 days at 32°. We used these YEA plates for two purposes: (1) we counted the colonies to assay the concentration of viable cells in the culture, and (2) we verified that the cultures contained diploids heterozygous for the correct markers via replica plating.

After at least 3 days, we isolated spores from the SPA plates as in (*Smith, 2009*). We then plated spores on YEA medium and grew the cells at 32° for 5 days. For viable spore yield fertility assays, we counted the number of colonies on the YEA plates and compared that number to the number of cells plated on SPA (*Smith, 2009*). In other words,

$$\text{viable spore yield} = \frac{\text{(colony forming units isolated from SPA [i.e., viable spores])}}{\text{(colony forming units plated on SPA)}}$$

It is important to note that the units of viable spore yields are not strictly the number of viable spores produced per diploid cell, so the values can be greater than four. We used the viable spore yield assay rather than micromanipulation of spores onto rich medium as our measure of fertility because the hybrids produced a large variety of spore morphologies that we were concerned could introduce unintended bias in spore selection. This variation was not a concern with the viable spore yield assay. We compared the *Sk/Sp* hybrid viable spore yields to the *Sk/Sk* values, rather than the *Sp/Sp* values, because the fold difference between *Sk/Sk* and the hybrids was similar when assayed via micromanipulation of random spores: ~24-fold when assayed via micromanipulation and ~25-fold when assayed via viable spore yield.

For the meiotic drive and recombination assays, we picked single colonies to master grids on YEA, incubated them them for 2 days and replica-plated them to supplemented YNB plates and YEA+drug plates to determine genotypes. We identified heterozygous diploid spores using codominant markers on either chromosome 1 or 2. We identified heterozygous aneuploid spores using codominant markers on chromosome 3. For example, in our *rec12Δ Sk/Sp* cross (SZY201xSZY208) we identified heterozygous diploids as His+ nourseothricin-resistant (*his5+/his5Δ::natMX4* ) cells. Similarly, we identified aneuploids as non-diploid Ade+ hygromycin-resistant (*ade6+/ade6Δ::hphMX4)* cells. We were unable to identify homozygous diploids or aneuploids using our markers and thus counted them amongst the haploids. We suspect such homozygous diploids and aneuploids were rare.

Recombination frequencies and genetic distances were calculated as in *Smith (2009)*. For the *Sp* genetic distances, we used the genome average of 0.16 cM per kb, as calculated in *Young et al. (2002)*. We used the *Sp* genome average because almost all of the genetic intervals are too large to be measured directly in *Sp*. For this reason, some of the actual genetic distances may be slightly larger or smaller than the values listed. Similar comparisons between the *Sk/Sp* hybrid and *Sk/Sk* recombination frequencies were not possible due to the lack of sufficient *Sk* genetic and sequence data. However, recombination calculations like those of *Young et al. (2002)* have not been carried out in *Sk* as this required an extensive catalog of selectable genetic markers built up by the *Sp* research community over many years.

## Genome-wide DSB frequency in *Sk/Sk* meiosis

We used the liquid sporulation protocol described above to induce meiosis in 1000 ml of *Sk/Sk rad50S, rec12-FLAG* diploids. Cells were harvested 8.5 hr after the induction of meiosis, and DNA covalently linked to Rec12 was purified. DSBs were mapped genome-wide using a *Sp* tiling array as in *Fowler et al. (2013)*. To account for incompatible probes, we compared the *Sk* dye signals to previously published *Sp* DSB arrays (*Fowler et al., 2013*) and filtered probes that had spuriously low signal in the input channel (values less than $10^{1.5}$). These probes likely represent genome positions either absent in *Sk* or that contain sufficient SNPs to interfere with hybridization. Hotspot intensities (*Figure 2A*) were then analyzed using 288 hotspot positions in *Sp* (*Fowler et al., 2013*) and integrating the median normalized Rec12 IP/input ratios from *Sp* and *Sk* at each hotspot. Two artificial hotspots not present in *Sk* but present in the *Sp* dataset were removed for clarity. Points on the x-axis of *Figure 2A* are greater than zero by definition, since the hotspots were determined in *Sp*. The correlation between hotspot intensities was quantified using the Pearson correlation coefficient ($r$). Raw and processed microarray data have been deposited in the NCBI Gene Expression Omnibus (accession no. GSE57039). Data analyses used R (http://www.r-project.org/) and Bioconductor (http://www.bioconductor.org/).

## Identification of the chromosome 1 inversion via whole-genome sequencing

We prepared *Sk* genomic DNA using the Genomic DNA Buffer Set (Qiagen) and Genomic-tip 20/G columns (Qiagen). Illumina sequencing libraries were prepared at the Fred Hutchinson Cancer Research Center core facility using Illumina TruSeq kits, and the libraries were sequenced in 100 base pair single-end, as well as 50 base pair paired-end, reads using an Illumina HiSeq 2000. All the sequence reads have been deposited in the NCBI sequence read archive (accession no. PRJNA244921). We assembled the single-end reads into contigs using the Velvet de novo assembler using a kmer value of 79 and default values for all other parameters (*Zerbino and Birney, 2008*). We used these contigs to screen out contaminant DNA sequences from the paired-end data set (this library was contaminated with non-yeast DNA) using Geneious software's 'map to reference' function using the contigs from single-end reads as the reference. We considered paired reads from the paired-end data set only when at least one of the two reads mapped to the reference. We then assembled these screened paired-end reads using Velvet default parameters except for kmer = 33 and insert size = 220. Only contigs longer than 1000 base pairs were analyzed further. We aligned the contigs to the *Sp* chromosomes using BLAST. We found two contigs that failed to align with one continuous piece of *Sp* chromosome 1. Instead, one end of the contigs aligned to one part of the chromosome and the other end aligned to a distant part in the opposite orientation. Both contigs had breakpoints at the same positions. We reasoned these could be due to inverted DNA in *Sk* relative to *Sp* and verified this using PCR.

## Identification of the reciprocal translocation breakpoints using whole-genome sequencing, Southern blots and PCR

We first identified the translocation via Southern blot hybridization (*Figure 4*). An additional Southern blot revealed that the translocated DNA did not include *ade7* on chromosome 2 (data not shown). We reasoned that we might be able to find the translocation junctions in a more complete *Sk* genome assembly utilizing long PacBio sequencing reads. PacBio sequencing libraries were prepared and sequenced at the University of Washington Genome Sciences core facility. These reads have been deposited in the NCBI sequence read archive (accession no. PRJNA244921). PacBio sequencing reads were used to join the contigs from paired-end reads into larger scaffolds using PacBio's SMRT Analysis software. We used AHA for the scaffolding using default parameters. None of the scaffolds showed evidence of the translocation found via Southern blot, which led us to assume the translocation

breakpoints were between two contigs that remained unjoined. We then designed PCR oligos near the ends of contigs that did not join in the regions where we predicted the translocation junction might lie (starting near the chromosome ends). PCR using oligos from unjoined contigs should form a product only if the contigs were in fact connected to each other. This strategy was successful (*Figure 4—figure supplement 5*).

## Pulsed-field gel electrophoresis and Southern blot analyses

Agarose DNA plugs were prepared and separated on pulsed-field gels as in *Hyppa and Smith (2009)*. For the Southern blot of DSB hotspots within the NotI restriction fragment D, we used a probe at the left end of the fragment known as probe c189 (chromosome 1 base pairs 1,024,213-1,025,212) (*Cromie et al., 2007*). For the Southern blot of the NotI restriction fragment J, we used probe c139 (chromosome 1 1,025,344-1,026,300) (*Cromie et al., 2007*). For the *ade6*, *leu1*, *alr2*, and *SPCP1E11.08* Southern blots, we used PCR products amplified either from within those genes or nearby using oligos listed in *Supplementary file 2*.

## Acknowledgements

We thank members of the Malik lab and D Barbash, N Elde, H Flores, N Johnson, C Peichel, N Phadnis and D Presgraves for comments on the manuscript and R Hyppa for technical training. This work was supported by a T32 HG000035 training grant and an American Cancer Society post-doctoral fellowship (to SEZ), an NIH grant R01 GM031693 (to GRS) and grants from the NIH, R01 GM074108, and the Mathers foundation (to HSM). HSM is an HHMI Investigator.

## Additional information

### Funding

| Funder | Grant reference number | Author |
| --- | --- | --- |
| National Institutes of Health | GM031693 | Gerald R Smith |
| Howard Hughes Medical Institute | | Harmit Singh Malik |
| G Harold and Leila Y. Mathers Foundation | | Harmit Singh Malik |
| American Cancer Society | | Sarah E Zanders |
| National Institutes of Health | GM074108 | Harmit Singh Malik |
| National Human Genome Research Institute | HG000035 | Sarah E Zanders |

The funders had no role in study design, data collection and interpretation, or the decision to submit the work for publication.

### Author contributions

SEZ, Conception and design, Acquisition of data, Analysis and interpretation of data, Drafting or revising the article, Contributed unpublished essential data or reagents; MTE, Conception and design, Acquisition of data, Analysis and interpretation of data, Drafting or revising the article; JSY, J-WK, KRF, Acquisition of data, Analysis and interpretation of data; GRS, Conception and design, Analysis and interpretation of data, Drafting or revising the article, Contributed unpublished essential data or reagents; HSM, Conception and design, Analysis and interpretation of data, Drafting or revising the article

## Additional files

### Supplementary files

• Supplementary file 1. Yeast strain description.

• Supplementary file 2. Oligo/primer description.

## Major datasets

The following datasets were generated:

| Author(s) | Year | Dataset title | Dataset ID and/or URL | Database, license, and accessibility information |
| --- | --- | --- | --- | --- |
| Zanders SE, Eickbush MT, Yu JS, Kang J, Fowler KR, Smith GR, Malik HS | 2014 | Schizosaccharomyces kambucha x Schizosaccharomyces pombe | PRJNA245039; http://www.ncbi.nlm.nih.gov/bioproject/PRJNA245039 | Publicly available at NCBI BioProject. |
| Zanders SE, Eickbush MT, Yu JS, Kang J, Fowler KR, Smith GR, Malik HS | 2014 | Schizosaccharomyces kambucha strain:SZY13 | PRJNA244921; http://www.ncbi.nlm.nih.gov/bioproject/PRJNA244921 | Publicly available at NCBI BioProject. |
| Zanders SE, Eickbush MT, Yu JS, Kang J, Fowler KR, Smith GR, Malik HS | 2014 | Genome Rearrangements and Pervasive Meiotic Drive Cause Hybrid Infertility in Fission Yeast | GSE57039; http://www.ncbi.nlm.nih.gov/geo/query/acc.cgi?acc=GSE57039 | Publicly available at the NCBI Gene Expression Omnibus. |

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
