## [Decision Letter]

Thank you for sending your work entitled “Genome Rearrangements and Pervasive Meiotic Drive Cause Hybrid Infertility in Fission Yeast” for consideration at *eLife*. Your article has been favorably reviewed by Detlef Weigel (Senior editor) and 2 additional external reviewers, one of whom, Kirsten Bomblies, has agreed to reveal her identity.

The three reviewers discussed their comments before we reached this decision and the Senior editor has assembled the following comments to help you prepare a revised submission.

The mechanisms that lead to speciation remain one of the most fascinating problems in modern biology. Genic incompatibilities, chromosomal rearrangements and genome doubling all provide avenues towards interspecific incompatibility. In this manuscript, a veritable tour de force, Zanders and colleagues dissect the genetic incompatibilities between the *Schizosaccharomyces pombe* (*Sp*) and *S. kambucha* (*Sk*) species, which differ in only 0.5% of their nuclear genome sequence. The authors go to great length, including the experimental analysis of DSB initiation hot spots, to ensure that sequence-based differences in recombination are not major contributors to incompatibility between the species. Instead, they find that despite the close sequence similarity, there are several major chromosome rearrangement that greatly reduce the fraction of viable spores. Similarly, the authors exclude that classic BDM incompatibilities are major contributors to interspecific incompatibility. Instead, they suggest that meiotic drive, with causal loci on each of the three chromosomes, is important. Provided that the latter conclusion holds up (see below), this study will speak to a wide readership, including scientists interested in speciation, unorthodox genetics, and the bizarre world of meiotic drivers.

It is remarkable that a species-barrier would involve three independent drivers. Since there appears to be one on each chromosome, this makes an extremely potent barrier to gene flow compared to most drive systems, where usually only a single chromosome is affected. Second, the translocation that switches two essential genes functionally spreads the effects of two of the drivers onto a different chromosome. This would likely be the first description among drivers of an effect on viability that necessitates the co-segregation of two distinct chromosomes from the driving genotype.

However, the reviewers were concerned whether the observations are indeed best explained by meiotic drive resulting in hybrid infertility, or rescue by aneuploidy overcoming already established hybrid infertility. The hallmark of a driver locus is that is increases its allele frequency faster than would be possible under classic Mendelian segregation combined with drift, but not due to selective forces (1:1 ratio in *Sp* x *Sk* crosses). Distorted Mendelian ratios in female meiosis are more easily assessed than in the exclusively male-like meiosis of *Schizosaccharomyces*. It is therefore difficult to discriminate between hybrid infertility/inviability and meiotic drive in male-like meiosis. For one, it is still unclear at what point during *Sp* x *Sk* crossing hybrid incompatibility occurs.

The increased ratio of aneuploidy (and diploidy) in *Sp/Sk* hybrids could indeed be due to meiotic drive, but there are alternative scenarios that you need to consider. Aneuploid tolerating mutants have been described in yeast (Torres et al 2010; Tange et al 2012). There is also the possibility of incompatibility between essential genes or loss of an essential gene due to the translocation, inversion or mutations (every 200th bp is different between *Sp* and *Sk*). Also, it cannot be ignored that dosage compensation could account for aneuploidy in *Sp* x *Sk* crosses. Finally, from both classical (B)DM models and Lenski's LTEE, a role for epistasis has emerged as a potential speciation factor. For example, the observed link between alleles on chromosomes 2 and 3 could be attributed to epistasis.

All considered, it seems premature to assign meiotic drive as the leading force of the observed distortion of allele ratios without addressing more directly any of the above mentioned established non-meiotic drive mechanisms. Ideally, this would of course be resolved by pinpointing the causal genes, but we would be happy to entertain other arguments.

---

## [Author Response]

The reviewers’ comments highlighted a possibility that we had considered but not elaborated, i.e., imposition and tolerance of aneuploidy may provide one explanation of the post-meiotic dysfunction we see in the *S. kambucha/S. pombe* (*Sk/Sp*) hybrids. We have increased the discussion of this possibility and clarified the distinction between traditional ‘meiotic drive’ and ‘post-meiotic dysfunction’ that is colloquially referred to as meiotic drive. Formally, aneuploidy susceptibility can be a means of imposing meiotic drive, but in additional evidence we present in our revisions, we show that this explanation cannot be the mechanism for the drivers we have uncovered on chromosome 1 and 2. Because chromosome 3 can tolerate aneuploidy, we cannot rule out aneuploidy as a mechanism for chromosome 3 drive. Although these drivers likely arose independently under a more typical scenario for spread of meiotic drivers, their joint action is what leads to the near complete sterility in the *Sp/Sk* hybrids.

We have used these comments to provide additional evidence and clarified explanations of our conclusion that meiotic drivers exist on all three *S. kambucha* chromosomes and significantly contribute to spore inviability in *Sk/Sp* hybrids. We address each of the comments in our point-by-point response to the reviewers. Although we agree that the ultimate identification of the drivers, and suppressors would reveal details about the molecular mechanisms behind this pervasive drive, we have found that the mapping is not straightforward due to chromosomal rearrangements on both *Sp* and *Sk* chromosomes, the possibility of epistasis and the lack of a high-throughput assay to assess meiotic success (in contrast to budding yeasts).

*[…] The reviewers were concerned whether the observations are indeed best explained by meiotic drive resulting in hybrid infertility, or rescue by aneuploidy overcoming already established hybrid infertility. The hallmark of a driver locus is that is increases its allele frequency faster than would be possible under classic Mendelian segregation combined with drift, but not due to selective forces (1:1 ratio in* Sp *x* Sk *crosses). Distorted Mendelian ratios in female meiosis are more easily assessed than in the exclusively male-like meiosis of* Schizosaccharomyces*. It is therefore difficult to discriminate between hybrid infertility/inviability and meiotic drive in male-like meiosis. For one, it is still unclear at what point during* Sp *x* Sk *crossing hybrid incompatibility occurs*.

We agree and have previously commented (Malik, TREE 2005) that the formal definition of meiotic drive as first proposed by [53] was limited to female meiotic drive. However, in later reviews, Sandler extended the term to include male meiotic drive (68): “Meiotic drive has been defined by Sandler & Novitski as any alteration of the normal process of meiosis with the consequence that a heterozygote for two genetic alternatives produces an effective gametic pool with an excess of one type; such a pattern of behavior will drastically alter the frequency of alleles in a population in such a way that a driven allele may increase in frequency in spite of deleterious physiological effects. This general concept has, however, been taken to include transmissional anomalies that are not strictly meiotic, but with similar populational consequences [see, for example, Lewontin]; this extended meaning seems justified and thus the more general definition will be used in this review.“

We now have a more extended discussion of examples of transmission distortion in male meiosis, and reference to the Sandler review.

We agree with the reviewers that in the *Sp/Sk* hybrid cross, decoupling the effects of the various chromosomal drivers complicates the interpretation of drive. This is why we believe our experiments in which we have separated the effects of the independent drivers are key (Figure 8 in revision). For instance, in the PKK x KKK cross, we observe an over-representation of the K1 chromosome of 65%, and a spore yield of roughly 80% of a conspecific KKK x KKK cross. Thus, K1 drive would fit the criteria required for a driver on chromosome 1 to increase in frequency by virtue of transmission distortion. We have similarly noted the ability for drivers on both chromosome 2 and 3 to independently drive, and meet the traditionally defined mathematical criteria for drive. We propose that each of these drivers arose independently or sequentially but it is their joint action (a circumstance under which drive did not originate) that leads to the near complete hybrid infertility, which would preclude the evolutionary success of either driver. Although we had a truncated discussion of this point, we have expanded this in our revision.

We agree that it is unclear at what point the spore death occurs in the *Sk/Sp* hybrids. We hope to address this question in future work identifying each of the loci required for meiotic drive.

*The increased ratio of aneuploidy (and diploidy) in* Sp/Sk *hybrids could indeed be due to meiotic drive, but there are alternative scenarios that you need to consider. Aneuploid tolerating mutants have been described in yeast (Torres et al 2010; Tange et al 2012). There is also the possibility of incompatibility between essential genes or loss of an essential gene due to the translocation, inversion or mutations (every 200th bp is different between* Sp *and* Sk*). Also, it cannot be ignored that dosage compensation could account for aneuploidy in* Sp *x* Sk *crosses. Finally, from both classical (B)DM models and Lenski's LTEE, a role for epistasis has emerged as a potential speciation factor. For example, the observed link between alleles on chromosomes 2 and 3 could be attributed to epistasis*.

We agree with the possibility that aneuploidy tolerance may be a means by which *Sk* chromosomes resist (and perhaps cause) drive and ultimately infertility. Indeed, were this to be the case, it would be a novel means of meiotic drive. However, it would also still be completely consistent with post-meiotic dysfunction in products of male meiosis, which have been likened to a toxin-antidote system in which the driver loci produce some toxin (here, aneuploidy-causing conditions) and suppressor loci protect against this toxin (herein, aneuploidy-tolerating mutations). Thus, although mechanistically unique, this would not preclude our conclusions of driver loci.

In response to reviewer suggestions, we have more formally considered in our revision whether aneuploidy causing and tolerating loci could explain the drive phenotypes we have observed. For this, we re-analyzed the chromosome 1 and 2 allele transmission data from the *rec12*^*+*^
*Sk/Sp* hybrid diploids, sorting them into two classes: aneuploid spores, and haploid spores. If the drive phenotypes of these two chromosomes we observed were related to the aneuploidy phenotype, the drive phenotype would be manifest primarily in the aneuploid spores. Instead, we observed strong drive of an allele on each chromosome in both the aneuploid and haploid spores. This experiment has been added to the text and the data are now displayed in Figure 6—figure supplement 2. In addition, the experiments presented in Figure 8 (Figure 7 in previous version) provide further support that the drive we observe of Sk chromosomes 1 and 2 (and the associated spore death) is independent of the aneuploidy phenotype. In these experiments we show that P1 K2 K3/K1 K2 K3 and K1 R2 K3/K1 K2 K3 diploids exhibit meiotic drive of K1 and K2, respectively. However, the level of aneuploid spores produced in these crosses is not higher than that in the Sk/Sk control. We have added text to the section describing these experiments to add the conclusion that the drive of Sk chromosomes 1 and 2 is not causally related to the aneuploidy phenotype. In contrast to chromosomes 1 and 2, we cannot robustly rule out that the aneuploidy phenotype may be related to drive on chromosome 3, partly because aneuploidy for chromosome 3 is viable in Sp and Sk.

It is important to note, however, that regardless of the mechanisms of drive or its exact relationship to aneuploidy, *Sk* chromosomes still have a significant transmission advantage over the *Sp* chromosome. And because these organisms undergo symmetric “male” meiosis to produce 4 gametes, the drive can occur only by causing the preferential death of spores that inherit the losing *Sp* alleles.

As the reviewers suggest, it is possible that some of hybrid genotypes are more tolerant of the stresses of aneuploidy, or that dosage compensation plays a role. We have now mentioned these possibilities in the Discussion. However, we do not feel that dosage per se provides the best explanation of our observed data. For example, we observe more than 7-fold more K1 K2 K3/P3 aneuploids (289) than K1 K2 K3 (32) haploids amongst the viable spores produced by *rec12Δ Sk/Sp* hybrids. K1 K2 K3 haploids represent the pure species genotype (and *Sk/Sk* diploids have high fertility), so it is hard to envision how adding P3 will provide a more than a 7-fold fitness boost, especially given the growth defects associated with aneuploidy. Rather, we favor the alternate model suggested by the reviewers that aneuploids are better equipped to survive the ‘toxic’ effects of drivers in hybrid meiosis.

We more formally address the possibility of classical BDM incompatibilities using two experiments. First, despite the fact that *Sp* chromosomes are intrinsically compatible, we recover the ‘pure’ *Sk* genotype ∼5X more frequently than the ‘pure’ *Sp* genotype from *rec12Δ Sk/Sp* diploids. The most parsimonious explanation for the low recovery of spores with the *Sp* genotype is meiotic drive.

Secondly, if there were indeed BDM incompatibilities that affected mitotic survival or fitness, we would expect impaired mitotic fitness of different haploid combinations of Sp and Sk chromosomes recovered from the *rec12Δ* hybrid cross, for instance, combinations of Sp chromosome 1 with *Sk* chromosomes 2 and 3. Contrary to this expectation, we find no perceptible growth defects associated with this ‘mis-assortment’ (new Figure 7). Other combinations are also equally viable. This implies that the reduction in recovery of *Sp* containing chromosomes is not due to an intrinsic mitotic defect; following meiosis, haploid combinations appear to be just as fit as conspecific meiotic products. In contrast, aneuploid cells grow slowly until they shed an extra chromosome.

Our experiments cannot rule out the possibility of BDM incompatibilities in *rec12+* crosses, as those would create a vast number of haploid combinations of *Sp* and *Sk* alleles, but the fact that the *Sp* and *Sk* chromosomes are largely mitotically compatible with each other and the fact that we still see drive and infertility in *rec12Δ* crosses whose products (except for chr 2-chr3 translocations we pointed out) are largely viable and fit, suggests that classical BDM incompatibilities are not the primary basis of speciation. This is in contrast to what has been found in other systems, including Drosophila, wherein BDM incompatibilities, dosage and epistasis play a key role in speciation.

However, our previous version failed to mention the important role that epistasis might play in *Sp/Sk* infertility. We have now remedied this problem and mentioned in the text that epistasis could, in addition to the chromosome translocation, be contributing to the genetic linkage we observe between alleles on chromosomes 2 and 3. We have now added the new Figure 7 and an extended discussion of this point in our revisions.

*All considered, it seems premature to assign meiotic drive as the leading force of the observed distortion of allele ratios without addressing more directly any of the above mentioned established non-meiotic drive mechanisms. Ideally, this would of course be resolved by pinpointing the causal genes, but we would be happy to entertain other arguments*.

We hope that our additional experiments and discussion would convince the reviewers that (13) two of the three chromosomal drivers act independent of aneuploidy, (33) that meiotic drive can act despite absence of genic incompatibilities, and (50) that each of the drivers indeed meets the classical definition of transmission distorters in male meiosis. We do agree with the reviewers that formal testing the mechanisms of drive will require the identification of the causal loci and potential suppressors. Unfortunately, those mapping experiments are beyond the scope of this paper, and are complicated by the chromosomal rearrangements and the possible role of epistasis. To put this mapping challenge in perspective, the three “Spore killer” meiotic drive elements in Neurospora, a fungus with roughly comparable genetic tools to fission yeasts, were identified over 30 years ago, yet their mechanism(s) of action are still unknown and the first critical gene was only recently identified. We hope that the several unique points of drive and speciation we have uncovered, the improvements we have made in our manuscript as well as the intriguing prospects of rampant, rapid speciation among fission yeasts, which is likely to spur further interest in speciation research in this system, would meet the editorial and reviewer expectations.